# Investigation of the Effect of Pumping Depth and Frequency of Flapping Hydrofoil on Suspended Matter Discharge Characteristics

Ertian Hua *, Mingwang Xiang, Tao Wang, Yabo Song, Caiju Lu and Qizong Sun

College of Mechanical Engineering, Zhejiang University of Technology, Hangzhou 310023, China;
211122020035@zjut.edu.cn (M.X.); 221122020431@zjut.edu.cn (T.W.); 221122020206@zjut.edu.cn (Y.S.);
221122020367@zjut.edu.cn (C.L.); 121222020046@zjut.edu.cn (Q.S.)
* Correspondence: het@zjut.edu.cn; Tel.: +86-135-8811-4369

**Abstract:** In order to study the effect of the pumping depth and pumping frequency of the flapping hydrofoil device on suspended solids in the waters, this paper takes raceway aquaculture as an example, and introduces a flapping hydrofoil device to improve the discharge of suspended solids in the raceway, in response to the problem of the deposition of suspended solids from fish faeces and bait residues in water. The CFD method was used to compare and analyze the discharge of suspended solids at different pumping depths, and the combined effect of the two was studied according to different combinations of pumping frequency and pumping depth. The results proved that the flapping hydrofoil motion can improve the bottom hydrodynamic insufficiency in ecological waters and thus enhance the discharge effect of suspended particles in water. In addition, the pumping depth of the flapping hydrofoil is too deep for the movement to be disturbed by the bottom surface, while the thrust generated by the flapping hydrofoil is weakened if the depth is too shallow. When the pump water depth is 1.1 H, the reversed Kármán vortex street is more stable under the balancing effect of the bottom surface and gravity, and the rate curve of the flapping hydrofoil acting on the discharge of suspended particles is better. From our comprehensive consideration of the joint effect of the pumping depth and pumping frequency, we recommend the use of a 1.1 H of pumping depth and 2.0 Hz pumping frequency in combination to achieve the best effect of discharging suspended particles. This study provides valuable insights into the actual engineering applications of flapping hydrofoil devices for improving water quality and ecological sustainability in raceway aquaculture.

**Keywords:** runway aquaculture; suspended particulate matter; flapping hydrofoil; pumping depth; pumping frequency

## 1. Introduction

The flapping hydrofoil device [1] is an underwater energy device that can increase hydrodynamics and improve the efficiency of discharge while protecting a fishery. In many ecological waters, insufficient hydrodynamics leads to the accumulation of suspended pollutants, exacerbating water quality degradation and water eutrophication [2]. To address the issue of inadequate hydrodynamics in ecological waters, a low-lift axial-flow pump [3] has previously been employed, but it encountered cavitation problems during practical usage [4]. Anderson et al. [5], utilizing the DPIV visualization of flow fields, discovered that the interaction between leading-edge and trailing-edge vortices generated during a flapping motion mimicking fish swimming produces counter-Kármán vortex streets, which effectively enhances the propulsion efficiency of the pumping at a low headlift. Despite possessing the significant advantages of low headlift and a high propulsion efficiency, flapping devices have not been widely used in practical pumping operations due to a lack of research on their application.

This paper addresses the issue of suspended particles in waters resulting from fish faeces and bait particles, using raceway aquaculture as an example. The use of a flapping

hydrofoil as a hydrodynamic source to improve the effluent effect of suspended particles in an aquaculture raceway is presented. Figure 1 shows a schematic diagram of a raceway aquaculture system; the raceway aquaculture technology is an innovative approach that replaces the traditional 'open free-range' pond mode with a 'captive' pond mode. This new model utilizes 'big ponds' (with an area proportion greater than 95%) to purify water and 'raceways' (with an area proportion less than 5%) to raise fish. It is a high-density aquaculture model that ensures the efficient use of space. However, a high aquaculture density will lead to the production of large amounts of nitrogen, phosphorus, and other farm wastes, as well as organic suspended particles during the aquaculture process. The accumulation of these wastes can inevitably lead to the deterioration of water quality, the spread of biological diseases, and a reduction in the benefits of aquaculture [6]. The organic suspended particulate matter in this instance is primarily composed of fish faeces and residual bait particles, such as humus and bacteria [7]. During the feeding process, approximately 25% of the feed is converted into suspended matter. The accumulation of fish faeces and bait residues can lead to calcification, decay, and the production of harmful substances such as ammonia and nitrogen. These substances will damage the health of the fish if they are not treated effectively [8].

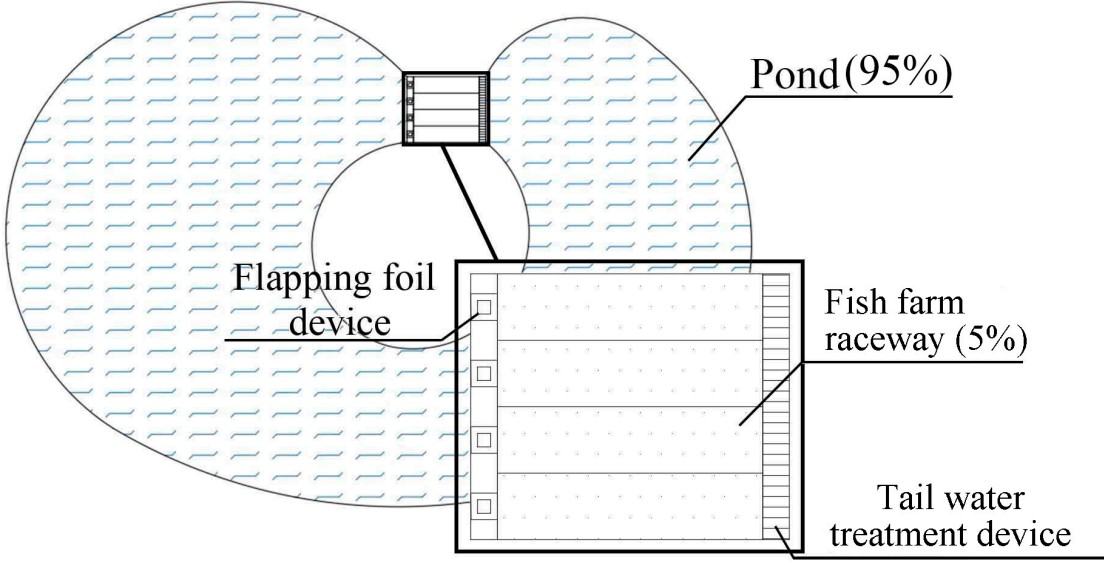

**Figure 1.** Schematic diagram of raceway aquaculture system.

To address the discharge issue in raceway aquaculture, Huggins et al. [9] employed CFD techniques to simulate the water flow and sediment transport within the flume. The study examined the effects of various sediment particle sizes on the settling efficiency, offering valuable insights into sediment distribution and flow patterns in raceway aquaculture. Brinker A. et al. [10] utilized a non-invasive laser particle sizing instrument to measure the particle distribution of suspended solids in raceway aquaculture. Their research uncovered the impact of the stocking density on the particle size distribution of suspended solids.

Overall, the unique structure of raceway aquaculture leads to various issues with effluent discharge. The current optimized design, which includes an aeration device [11], has hydrodynamic limitations and creates a reflux zone at the inlet, leading to particle accumulation. This paper introduces a bionic flapping hydrofoil device [12] that can effectively address the issues of insufficient water dynamics and particle accumulation in the reflux zone, thereby improving the hydrodynamic situation of the runway. Additionally, a new raceway aquaculture model of runway push water can be formed by incorporating a sinking collector as a tail water treatment system at the end of the runway. A previous study verified that the aquaculture model is more efficient than the existing raceway aquaculture model [13].

Regarding the flapping hydrofoil device, R Knoller [14] found that the angle of attack of flapping hydrofoils generates a normal force component in the positive direction. This discovery proves the feasibility of using flapping hydrofoils for propulsion and pumping applications. Currently, research is mainly focused on the kinematic and structural parameters of flapping hydrofoil. Ding et al. [15] developed a model to estimate the efficiency of underwater flapping hydrofoil propulsion. They analyzed the relationship between the flapping frequency, lift amplitude, pitch amplitude, and other kinematic parameters relating to the flapping hydrofoil thrust. They obtained the change curve of the underwater flapping hydrofoil propulsion efficiency with the kinematic parameters and the optimal point of propulsion efficiency. This study provides a significant contribution to the understanding of flapping hydrofoil kinematic parameters. Du et al. [16] conducted a comparison of the propulsive performances of flapping hydrofoils using four common flapping modes. They established a two-degree-of-freedom underwater rigid flapping hydrofoil computational model, which provides an effective reference for selecting the appropriate flapping mode. The study demonstrated that the positive arc-shaped oscillating mode had superior hydrodynamic characteristics compared to the linear and negative arc-shaped oscillating modes. Xiao et al. [17] compared the nonlinear correlations of the modal coefficients of flapping hydrofoils of different types by constructing a degradation model. The study confirmed the unity of the nonlinear correlations of bionic wing types. These findings provide an important reference for an in-depth understanding of flapping hydrofoil hydrodynamic characteristics.

The aim of this study is to conduct an in-depth analysis of the effect of flapping hydrofoil devices on the discharge of suspended particles such as feed residues and fish faeces in aquaculture raceways under different pumping depths and frequencies. This analysis was achieved through numerical simulations using the commercial computational fluid dynamics (CFD) simulation software Ansys-Fluent-2020R2. In this research, the k-ε turbulence model was selected to accurately capture turbulence effects. The simulation results were validated by comparing them with experimental data, ensuring the reliability and accuracy of both the discrete and continuous phase models. This validation approach involved a comprehensive assessment of the model's performance in predicting both discrete and continuous phases against experimental data. Furthermore, this study investigated the optimization configuration of different pumping depths and frequencies to provide valuable insights for enhancing the effluent performance of raceway aquaculture systems.

## 2. Modelling

### 2.1. Description of Flapping Hydrofoil Motion

As shown in Figure 2 in this study, a NACA0012 airfoil with a chord length (c) c = 1 m was used to pump water, with the lift and pitch motion as the power source. According to the previous research on flapping hydrofoil pivot positions [18], the fixed flapping hydrofoil pitch pivot position of 0.2 c will obtain the optimal flapping hydrofoil water push efficiency. In order to ensure the practical applicability of this paper, this paper takes the 0.2 c pivot position as an example, so that the flapping hydrofoil can achieve the optimal hydrodynamic performance as much as possible in motion.

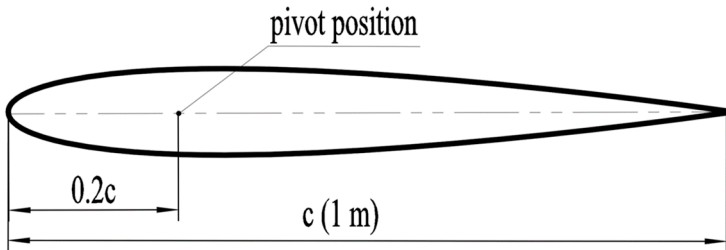

**Figure 2.** Schematic diagram of NACA0012 flapping hydrofoil structure.

For the flapping hydrofoil motion mode, to reduce the complexity of the mechanism's physical design, this paper takes the linear sinusoidal lift and pitch motion commonly used in flapping hydrofoils as an example, for which the basic motion equations are as follows:

$$h(t) = h_{max} sin(2\pi ft), \tag{1}$$

$$\theta(t) = \theta_{max} sin(2\pi ft + \varphi), \tag{2}$$

where $\theta_{max}$ is the pitch amplitude. According to the research of Ding et al. [15], the propulsive efficiency of a flapping hydrofoil starts to decline after $\theta_{max}$ exceeds $\pi/6$ rad; therefore, in order to improve the hydrodynamic force of flapping hydrofoil in practical use, this paper takes $\theta_{max} = \pi/6$ rad. $h_{max}$ is the lift and sink amplitude; this paper, according to the chord length of the flapping hydrofoil, c, takes $h_{max}$ to be 0.5 c; $\varphi$ is the phase difference of lift and sink with the pitch amplitude. The flapping frequency, *f*, is the object of study in this paper, and the specific parameter determinations are mentioned in the results.

To ensure the optimal hydrodynamic performance of the flapping hydrofoil as much as possible, based on previous research, phase adjustment is introduced to optimize flapping motion mode three in reference [19], which serves as the flapping hydrofoil motion model in this paper. The optimized flapping hydrofoil motion equation is as follows:

$$h(t) = h_{max} cos(2\pi ft), \tag{3}$$

$$\theta(t) = \theta_{max} sin(2\pi ft), \tag{4}$$

Figure 3 shows the optimized flapping hydrofoil motion, adjusted to prevent the upward offset of the reversed Kármán vortex street caused by the tilted initial pitch angle, thereby improving its stability. This also avoids positional errors caused by tilt angles during meshing. Furthermore, in practice, using the limit position of the reciprocating motion as the initial position of the motion is more reasonable and makes it easier to design and control the mechanism.

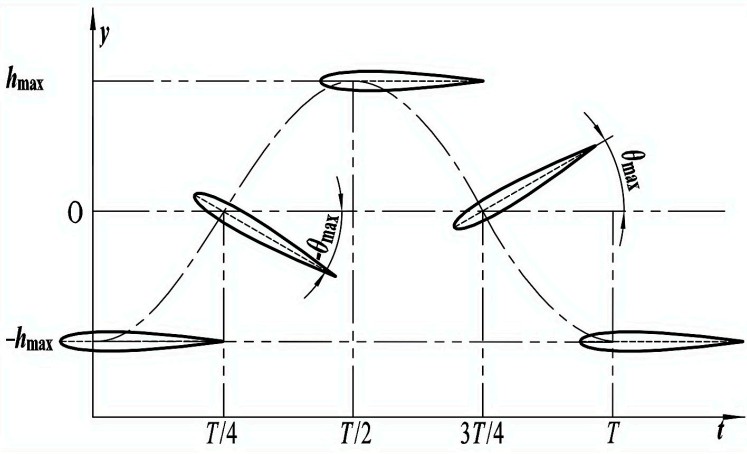

**Figure 3.** Schematic Diagram of flapping hydrofoil lift and pitch motion.

### 2.2. Particle Tracking Models and Discrete Phase Parameters

This study utilizes the commercial software Ansys-Fluent to establish a simulation model. The fluent discrete phase model (DPM) is adopted to simulate the drift and settlement characteristics of suspended particles under a flapping hydrofoil. The DPM defines the fluid as a continuous phase and the particulate matter as a discrete phase, and describes the motion of the discrete particles based on the Lagrangian method. In this study, the effect of the particles on the fluid has been ignored due to their small size and mass. To predict the particle motion, the integral form of the differential equation of motion acting

on the particles in Lagrange coordinates is used to obtain particle trajectories. The solid particle position equations and the differential equations of the forces are as follows [20]:

$$\frac{d\mathbf{u}_p}{dt} = F_D(\mathbf{u} - \mathbf{u}_p) + \frac{g_x(\rho_p - \rho)}{\rho_p} + F_X G_k, \tag{5}$$

where $u_p$ is the velocity vector of the particles, in m/s; $\rho_p$ is the density of the particles, in kg/m$^3$; $F_D(\boldsymbol{u} - \boldsymbol{u}_p)$ is the Traction force per unit mass of the particles, in N; $g_x(\rho_p - \rho)/\rho_p$ is the gravitational force per unit mass of the particles, in N; and $F_X$ denotes the component force of the other forces in the X-direction, in N. This study focuses solely on the particle drift-settling characteristics and does not consider the evaporation and dissolution of the particles, nor does it consider the energy balance equation of the particles in the CFD calculation.

In computational fluid dynamics (CFD) calculations, the main parameters of particles in the discrete phase are divided into particle density and particle size, as well as particle mass flow rate. To ensure that the simulation closely resembles reality, this paper will determine the particle parameters based on existing research and actual measured particle data on high-density aquaculture. Using sinking feed and fish faeces from common cold water predatory fish in raceway aquaculture as an example [21], this study selected a density of 1100 kg/m$^3$, which is the average value of sinking particles, and the particle diameter is assumed to be the average diameter, $d_p$ = 0.5 mm, with a certain degree of particle size distribution. In the simulation model, the fluctuation range of the particle size distribution is about ±0.05 mm. It is notable that fish faeces particles may decompose into smaller particles over time, thus changing their diameter, and the effect of this on the results of this study is manifested in the change in the escape rate of the particles. However, this decomposition process is obviously detrimental to the applicability of the findings of this paper because this stochastic process does not occur in a stable form; therefore, in order to ensure the accuracy of the results of this study and to avoid errors due to the stochastic decomposition of the particles, particle decomposition was ignored.

According to Zhang et al. [22], the concentration of total particulate matter (TPM) of the suspended particles of high-density aquaculture water was 31.58 ± 3.38 mg/L for residual bait and 29.75 ± 5.26 mg/L for fish faeces. The runway dimensions used in this study were 20 m × 5 m × 2 m, with a water volume of approximately 200 m$^3$. Based on this volume, the total mass of the particles was calculated to be 0.6316 kg (residual bait) + 0.6612 kg (fish faeces), resulting in a total of 1.2266 kg. To determine the mass flow rate, it is necessary to estimate the settling velocity of the particles.

In real conditions, particles of different shapes will show different motion characteristics in water, such as oval-shaped particles which will produce a continuous rocking motion during the settling process. The effect of the particle shape on the velocity of the particle movement is particularly obvious in the case of direct frictional contact with the wall. In this study, the particles form deposits after contacting the bottom, making them close to stationary, which makes the effect of the shape of the particles on the results of this study significantly reduced. Therefore, in order to facilitate this study's calculations, the particle shapes are all simplified to spherical in this paper. According to the literature [23], the settling velocity of particles in a Newtonian fluid can be calculated based on the corresponding Reynolds number interval. We define the particle settling Reynolds number as:

$$\mathrm{Re} = \frac{d_p \cdot \rho_f \cdot v_f}{\eta}, \tag{6}$$

where $\eta$ is the viscosity of the liquid, which, for water, is $1.005 \times 10^{-3}$ Pa·s; $v_f$ is the raceway inflow velocity, which is in the range of 0.4–1.2 m/s; $\rho_f$ is the density of water, which is 998.2 kg/m$^3$; $d_p$ is the particle diameter, which is 0.5 mm = $5 \times 10^{-4}$ m. Using the above equation, we obtain Re = 198.64–595.94 for the particle phase in this paper.

Within this Reynolds number interval, the particle settling velocity is determined by the following equation:

$$v_p = d_p \sqrt[3]{\left(\frac{\rho_p - \rho_f}{\rho_f} \frac{2g}{15}\right)^2 \frac{\rho_f}{\eta}},$$ (7)

where $v_p$ is the settling velocity of particles, in m/s; $\rho_f$ is the density of the liquid, which, for water, is taken as 998.2 kg/m$^3$. After substituting the above equation into the parameters of this paper, the particle settling velocity $v_p$ is obtained as $2.54 \times 10^{-2}$ m/s. Notice here that the drag coefficient is not involved in Equation (7), but in the high Reynolds number interval, the velocity of particle motion is affected by the drag coefficient, and the equations for particle settling velocities in the high Reynolds number range are given in the literature [23]. According to the settling velocity, the mass flow rate of the particles can be further calculated, and the mass flow rate equation is:

$$Q = \rho \cdot A \cdot v_p,$$ (8)

where $Q$ is the mass flow rate, in kg/s; $\rho$ is the density of particles, in kg/m$^3$; $A$ is the flow cross-sectional area, which takes the value of 2 m $\times$ 5 m = 10 m$^2$. After substituting the particle settling velocity, the mass flow rate $Q$ is obtained as 280.39 kg/s.

### 2.3. Evaluation of Indicators and Particle Placement Strategy

The raceway aquaculture model purifies wastewater by adding a tailwater treatment device at the end of the raceway. The treated wastewater is then discharged into the reservoir. This paper uses the sinking collector device as an example. The device collects discharged particles through the sinking collector tank and then sucks them out through the suction device to form a complete tail water purification system. To investigate the impact of suspended particle discharge, the particle escape rate is typically used as a measure. The particle escape rate is calculated as follows:

$$E_R = \frac{N_E}{N_f + N_E} \times 100\%,$$ (9)

where $E_R$ is the particle escape rate, $N_E$ is the total particle escape, $N_f$ is the total number of particles in the fluid domain. Both the flapping hydrofoil and particle motions are carried out in a simplified two-dimensional rectangular runway basin. To better match the actual raceway aquaculture runway, the runway is 21 m long, the water depth is 2 m, and the inlet is 1 m away from the flapping hydrofoil position. Figure 4 shows the pumping depth, which is defined as the distance between the center position of the flapping hydrofoil's lifting and sinking motion and the water surface. In the subsequent study, the pumping depth was measured using the indicator H = 1.0 m, which ranged from 0.8 H to 1.2 H.

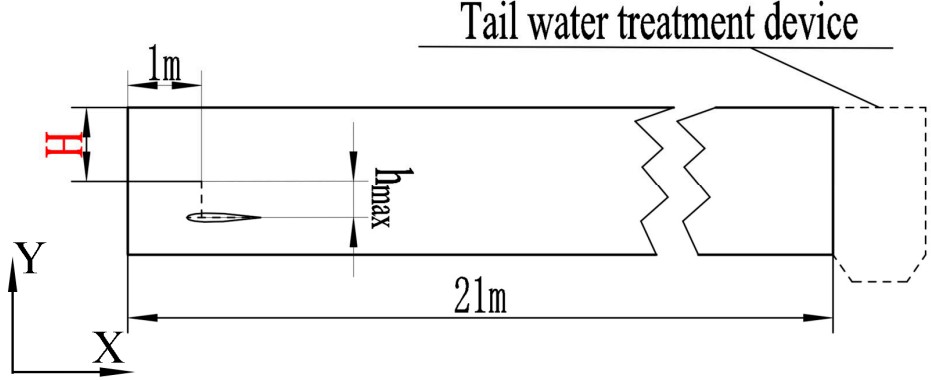

**Figure 4.** Schematic diagram of the global structure.

To comprehensively capture the deposition and discharge process of particles in ecological water bodies and reduce computational errors caused by water–air interface tension, this paper established 15 particle injection points uniformly at 0.5 m from the water surface, ranging from 4 m to 19 m from the inlet. The diameter of the particle cluster injection was 0.1 m. The particle injection points are shown in Figure 5. To ensure consistency, the particle injection time and the total calculation time should be the same, and non-constant particle tracking should be used.

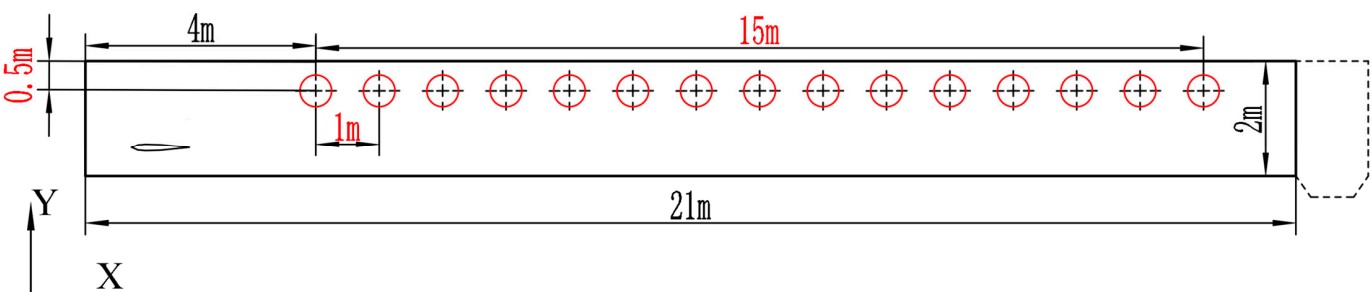

**Figure 5.** Schematic diagram of pellet placement strategy.

This paper analyses the discharge performance of flapping hydrofoil under different pumping depths using the escape rates and rate curves of particles as indicators. The formation mechanism of the escape rate and rate curve is explained through the flow field structure at different pumping depths. By analyzing the escape rate curves under the coupling of different pumping frequencies and depths, the joint effect of the pumping depth and frequency of the flapping hydrofoil on the discharging performance of suspended particles is investigated, and the reference value of the parameter coupling under different discharging requirements is finally given according to the results.

### 3. Numerical Method and Validation

#### 3.1. Control Equations and Turbulence Modelling

This paper presents CFD simulations carried out using the commercial software Ansys-Fluent. All paid features of the software have been licensed. For the incompressible flow turbulence problem, the equations of the motion control are the Reynolds-averaged Navier–Stokes (RANS) equations and the time-averaged continuity equations [24]:

$$\frac{\partial \boldsymbol{u}_i}{\partial \boldsymbol{x}_i} = 0, \tag{10}$$

$$\frac{\partial \boldsymbol{u}_i}{\partial t} + \boldsymbol{u}_j \frac{\partial \boldsymbol{u}_i}{\partial \boldsymbol{x}_j} = -\frac{\partial p}{\partial \boldsymbol{x}_i} + \frac{\partial}{\partial \boldsymbol{x}_j}\left[(\gamma + \gamma_t)\left(\frac{\partial \boldsymbol{u}_i}{\partial \boldsymbol{x}_j} + \frac{\partial \boldsymbol{u}_j}{\partial \boldsymbol{x}_i}\right)\right], \tag{11}$$

where $\boldsymbol{u}_i$, $\boldsymbol{u}_j$ ($i, j$ = 1, 2) are the average velocity of fluid movement in the $i$, $j$ direction, in m/s; $\boldsymbol{x}_i$, $\boldsymbol{x}_j$ ($i, j$ = 1, 2) are the spatial coordinates in the $i$, $j$ direction, in m; $p$ is the fluid pressure, in Pa; $\gamma$ is the laminar viscosity coefficient, in Pa·s; $\gamma_t$ is the turbulence viscosity coefficient, in Pa·s. The turbulence viscosity coefficient, $\gamma_t$, is calculated as:

$$\gamma_t = C_\mu \frac{k^2}{\varepsilon} \tag{12}$$

where $k$ is the turbulent kinetic energy, J; $\varepsilon$ is the turbulent kinetic energy dissipation rate, in m$^2$/s$^3$; $C_\mu$ is the dimensionless constant. Various turbulence models are provided in Ansys-Fluent for simulating various fluid flow problems. Among the commonly used RANS turbulence models such as the Spalart–Allmaras, Reynolds Stress, and k-$\varepsilon$, k-$\omega$ models, which are based on the equations of turbulent kinetic energy and turbulent dissipation rate. However, the computational model of Spalart–Allmaras is not widely tested and lacks

sub-models; the Reynolds Stress Model occupies a larger computational volume and is more difficult to converge; and the k-ω model is more difficult to converge and requires more time compared to the k-ε model. Therefore, the k-ε turbulence model is chosen for the computation in this paper. Fluent provides several versions of the k-ε model, including the standard k-ε model, RNG k-ε model, and Realizable k-ε model, among which the standard k-ε model may have some limitations when dealing with the wall boundary layer, and the RNG k-ε model is more computationally intensive. The Realizable k-ε model is more accurate in dealing with turbulence anisotropy and wall boundary layers compared to the standard k-ε model, and it improves the reliability of the model by taking into account the interrelationships among the turbulence statistics. Therefore, the Realizable k-ε turbulence model is used to solve the N–S equation in this paper. The equations can be found in the literature [24].

*3.2. Computational Domain and Meshing*

Since three-dimensional computation would require exponentially more computational resources and would have no effect on the results, the simulation and computational models in this paper are in two dimensions, and the appropriateness of the dimensionality will be explained later in the validation of the methodology. When performing motion simulation, the overlapping mesh method is commonly used to avoid the negative volume problem caused by large mesh changes. However, this method is incompatible with the DPM discrete phase model in CFD simulation. Therefore, this paper uses the dynamic mesh method to simulate the underwater motion of flapping hydrofoil. The specific boundary conditions are set as shown in Figure 6; in order to more realistically simulate the boundary flow around the flapping hydrofoil and the wall, and to reduce the numerical calculation error, the boundary layer is divided around the edge of the flapping hydrofoil as well as the bottom surface, and the thickness of the first layer of the mesh is 0.0001 m (y+ = 1).

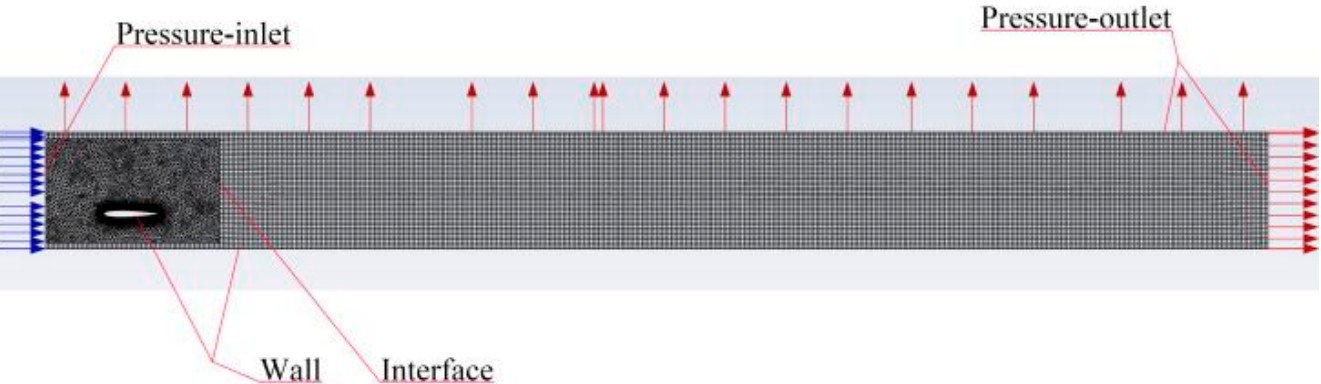

**Figure 6.** Schematic diagram of boundary condition settings.

Figure 7 shows the meshing of the computational domain and the initial moment. To ensure the mesh quality and avoid a negative volume, the airfoil boundary layer and outer mesh are separated into a static domain that is not involved in mesh reconstruction. Unstructured meshing is used in the dynamic mesh region.

The motion grid is established using the diffusion smooth method combined with local cell grid reconstruction. The diffusion coefficient is defined as 3. For local cell grid reconstruction, the maximum cell skewness is set to 0.6, and the grid is re-divided at each iteration step. The motion of the flapping hydrofoil is defined by user-defined directives (UDF). The motion mesh at different moments within a single cycle is shown in Figure 8. The solution employs the Coupled algorithm to couple the pressure and velocity fields and uses the second order upwind (SOU) format for discrete time.

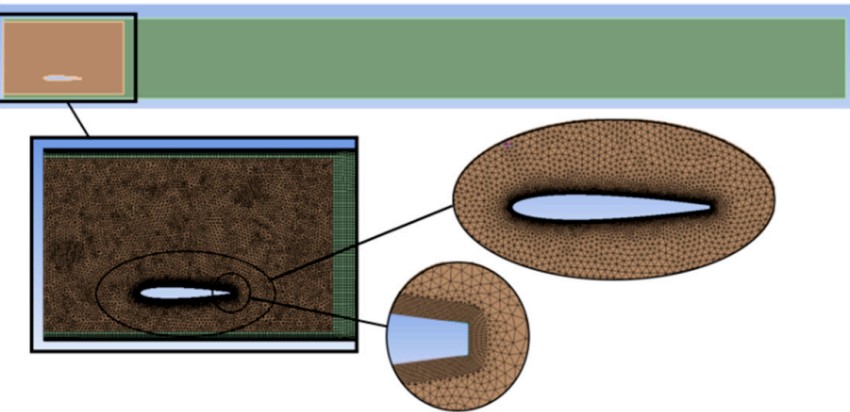

**Figure 7.** Schematic diagram of grid division.

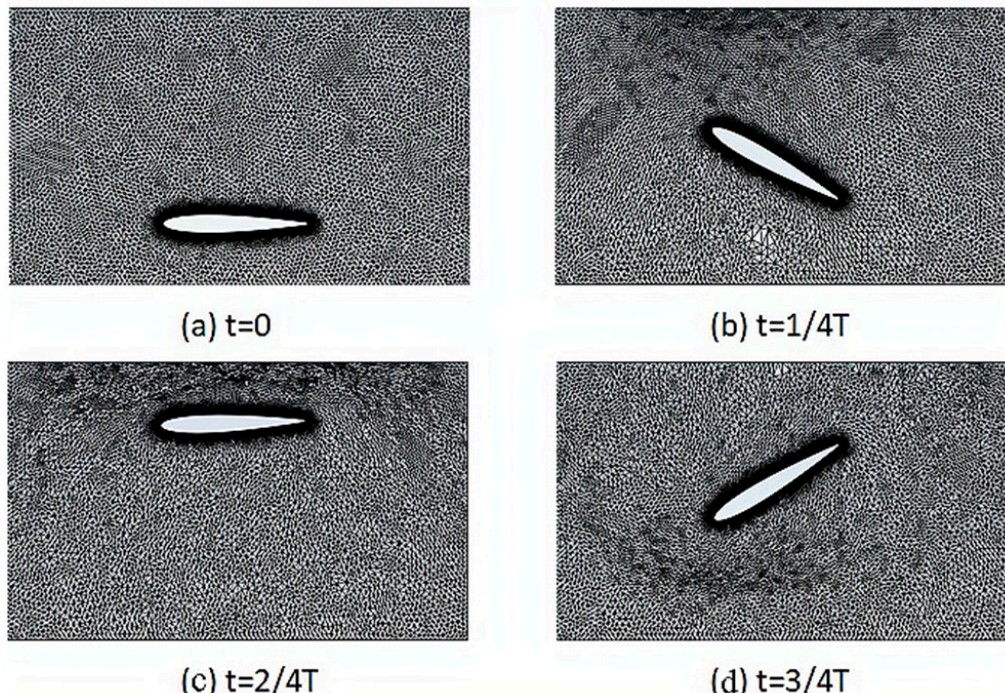

**Figure 8.** Schematic representation of grid changes during four major periods of a cycle.

### 3.3. Grid-Independent Verification

In the simulation calculation in this study, the dual AMD EPYC 7542 32-Core Processor is used, for which the running memory is 256 G. To enhance the computational efficiency while ensuring accuracy, we verified the irrelevance of the grid number and used different grid numbers for the simulation calculations. Figure 9a shows the grid-independence validation diagram, which compares the change curves of the flapping hydrofoil thrust in a stable flow field under different grid numbers; we observed significant errors in the model when using 15,981 grid numbers. After comparing the flapping hydrofoil thrust under the 95,199 and 150,911 grids, it was found that the calculation results were not affected by the improvement of the grid number. Therefore, the final grid number of 95,199 was chosen. In the frequency range from 0.5 Hz to 1.5 Hz, this paper adopts a time step of 0.01 s and sets the criterion of convergence within $1 \times 10^{-5}$, and the computation time consumed in this condition is about 5 h. In the computation, a negative volume situation of the moving mesh occurs frequently as the frequency increases, and a negative volume of the dynamic mesh occurs frequently as well. In order to improve the negative volume problem, a time step of 0.005 s is used at 1.75 Hz and 2 Hz, and the number of iteration steps is adjusted so that the overall time remains unchanged, which increases the computational time by

a factor of about two. For 2.5 Hz and 3 Hz, a time step of 0.001 s is used. Comparing the convergences at different time steps, it was found that there is no difference between the two. In order to ensure that the time step does not affect the results of this calculation, this paper compares the change curves of the flap thrust coefficients under the time steps of 0.01 s, 0.005 s, and 0.001 s, respectively, as shown in Figure 9b, and it is obvious that the influence of the selected time step on the calculation error is relatively small and can be ignored.

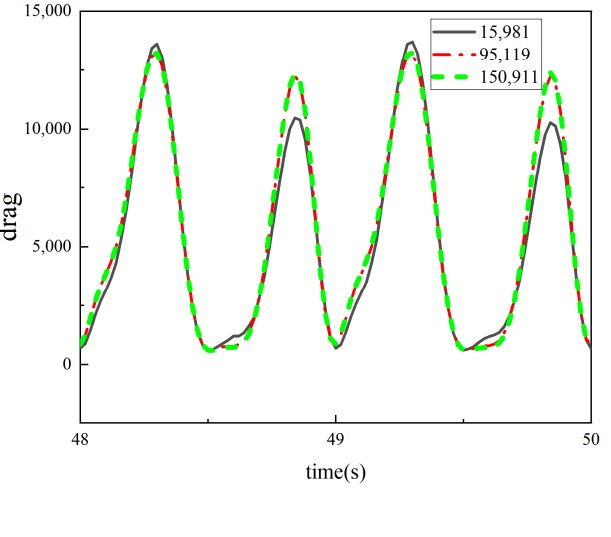

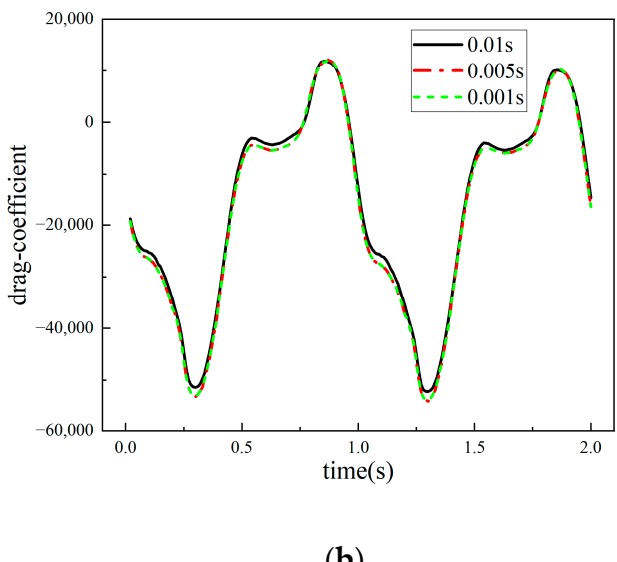

**(a)**

**(b)**

**Figure 9.** Verification of grid-independence and time-step-independence. (**a**) Thrust curves of a flapping hydrofoil in a stable flow field with varying grid numbers. (**b**) Thrust coefficient curves of flapping hydrofoils with different time steps.

### 3.4. Validation

To ensure the accuracy of the two-dimensional model results, this paper carries out numerical simulations of both discrete and continuous phases. This is done to verify the numerical simulation in two dimensions and prevent large discrepancies between the numerical calculation results and the actual results. The settling velocity of quartz sand particles in clear water was calculated using the discrete phase model, based on the cylinder model test described in the literature [23]. Figure 10 shows the change curve of the settling velocity with respect to the particle size. As the density increases, the settling velocity of the particles will inevitably increase, and, thus, the calculation error will also increase. Therefore, the accuracy of the calculation of the particle phase at higher densities can ensure that the results of the discrete phase calculation at smaller densities are closer to the real situation. The density of quartz sand is much higher than the density of the 1100 kg/m$^3$ particles used in this paper, and for the discrete phase model used in this paper in the simulation calculation of quartz sand, the calculation errors are kept within 0.0316 m/s, which is sufficient to prove the reliability of the method used in the study.

To verify the accuracy of the continuous phase calculations, a comparison was made with experimental data from the literature [25]. A two-dimensional NACA0012 hydrofoil with a chord length of c = 0.1 m was used. The incoming velocity was 0.4 m/s, with a lift and sink amplitude of $h_{max}$ = 0.075 m, and a maximum angle of attack of 20°. Upon observation of Figure 11 and comparison of the thrust coefficients at different Strouhal numbers (St), it is evident that the calculated values in this paper align well with the experimental values. Therefore, the numerical model adopted in this paper meets the required level of calculation accuracy.

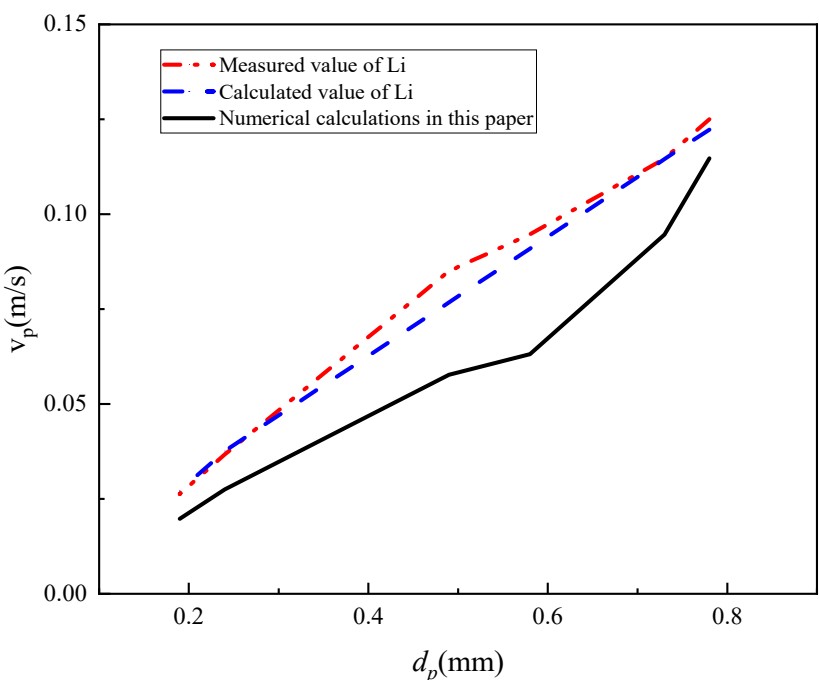

**Figure 10.** Comparison of simulation and experimental results on the variation curves of settling velocity with grain size for quartz sand with different grain sizes.

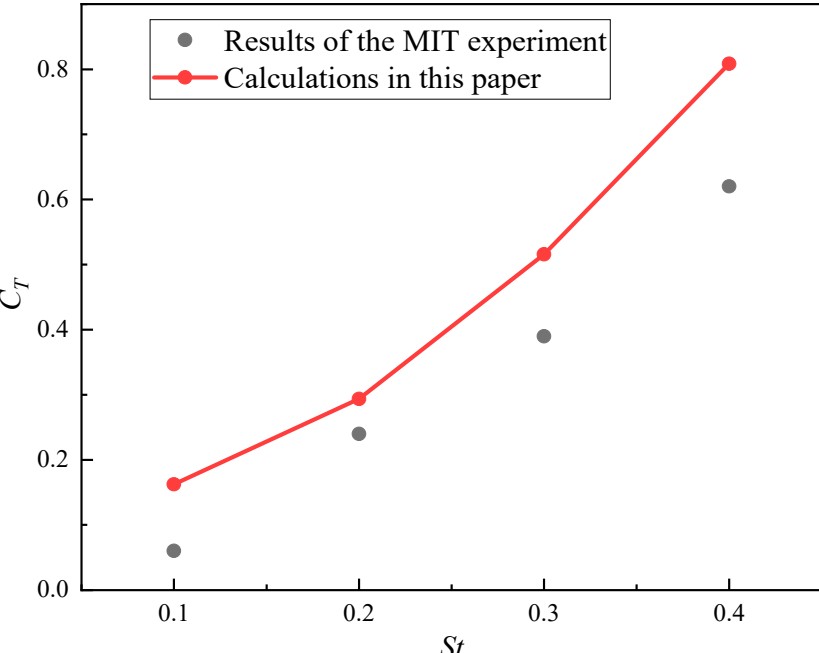

**Figure 11.** Comparison of numerical and experimental results for the variation in flapping hydrofoil thrust coefficient at different Strouhal numbers.

## 4. Results & Discussion

### 4.1. Impact of Flapping Hydrofoil Motion on the Distribution of Suspended Particles in Water

To investigate the impact of flapping hydrofoil water pumping on the discharge of suspended particulate matter, the particle distributions at different given times were selected for comparison, from the beginning of the particle introduction into the water to the basic discharge of the flow channel. The pumping depth was fixed at 1.0 H and the pumping frequency at 1 Hz to facilitate this study. The particle distribution over time was recorded from the steady state of the flow field. By calculation of the velocity vectors at

each point in the steady flow field, the velocity nephogram of the flow field was obtained, as shown in Figure 12.

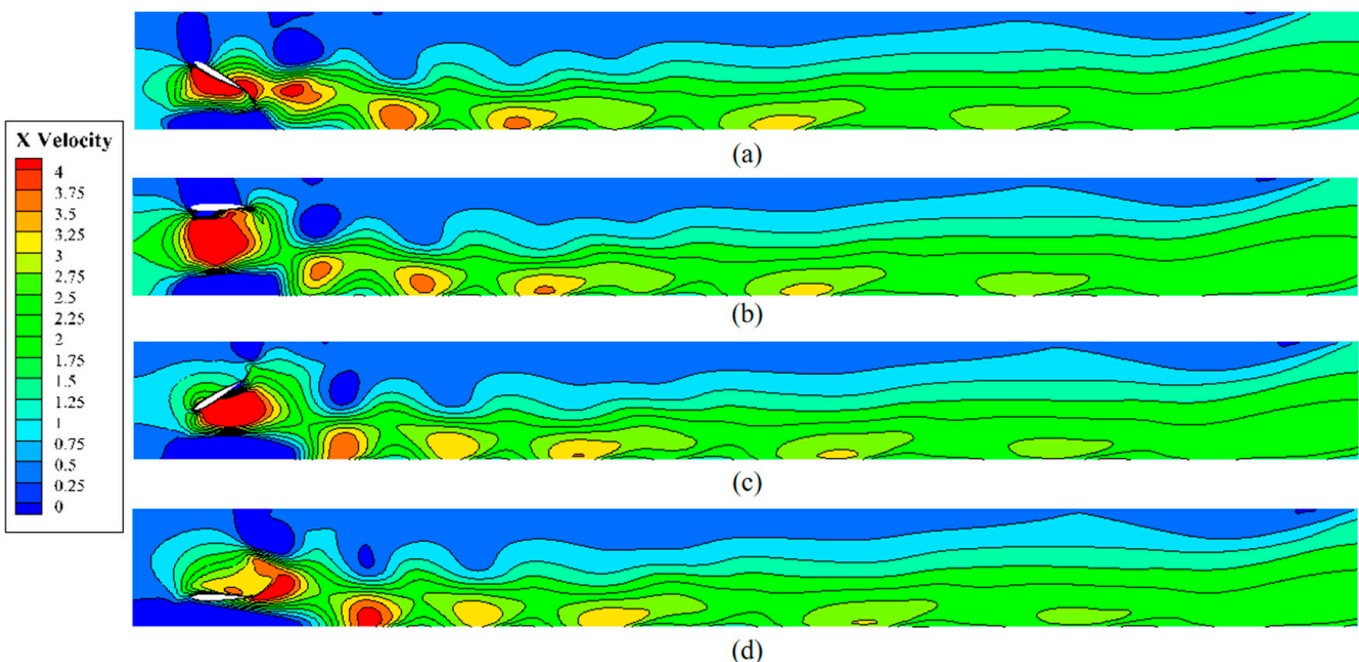

**Figure 12.** Velocity nephogram of the flow field during a single cycle at steady state: (**a**) 1/4 T, (**b**) 1/2 T, (**c**) 3/4 T, (**d**) 1 T.

The vortex nephogram shown in Figure 13 was obtained by calculating the vorticity of the velocity vectors at each point in the steady flow field. According to Figure 13, a pair of anisotropic anti-Carmen vortices is generated by the lift-sink-pitch motion in a single cycle when the flapping hydrofoil pumps water. The reversed Kármán vortex street refers to the alternating pattern of swirling vortices shed by a bluff body, such as a flapping hydrofoil, in a fluid flow. These vortices play a crucial role in enhancing the propulsion efficiency by generating thrust and promoting fluid mixing. In the context of our study, the stability and characteristics of the reversed Kármán vortex street directly influence the discharge rate of suspended particles in raceway aquaculture systems. By optimizing the pumping depth, we aim to maximize the stability of the reversed Kármán vortex street, thereby improving the propulsion efficiency of the flapping hydrofoil device and enhancing the discharge of the suspended particles. Therefore, understanding the physical principles underlying the reversed Kármán vortex street is fundamental for optimizing the performance of flapping hydrofoil devices in raceway aquaculture applications. This study employed the post-processing of simulation results to examine and visualize the reverse Kármán vortex streets. This post-processing procedure involved analyzing and visualizing the simulation results of the flow field, including the formation and characteristics of the vortex streets. Various techniques such as contour plots, vector plots, and streamline visualization were utilized to illustrate the behavior of the vortex streets.

In the steady state of the flow field, the flapping hydrofoil generates a reversed Kármán vortex street in the water, which gradually decreases and dissipates under the influence of gravity, and the vortex strength decreases with increasing distance from the flapping hydrofoil and evolves into a stable stratified flow velocity after dissipation. In addition, the reversed Kármán vortex street is dissipated at the bottom surface, resulting in a relatively larger vortex near the bottom surface, so that the flow velocity near the bottom surface overcomes some of the shear stresses and does not decay significantly, which can effectively inhibit the particle deposition phenomenon. In Figure 13, it can be observed that no obvious backflow phenomenon was generated in the raceway during the flow process. This result

should laterally prove the potential of flapping hydrofoil for discharge in an open channel or square pipe, but whether it has the same effect in round pipes is still to be discussed due to the limitation of the model dimensions and our ignoring the change in the z-direction flow field.

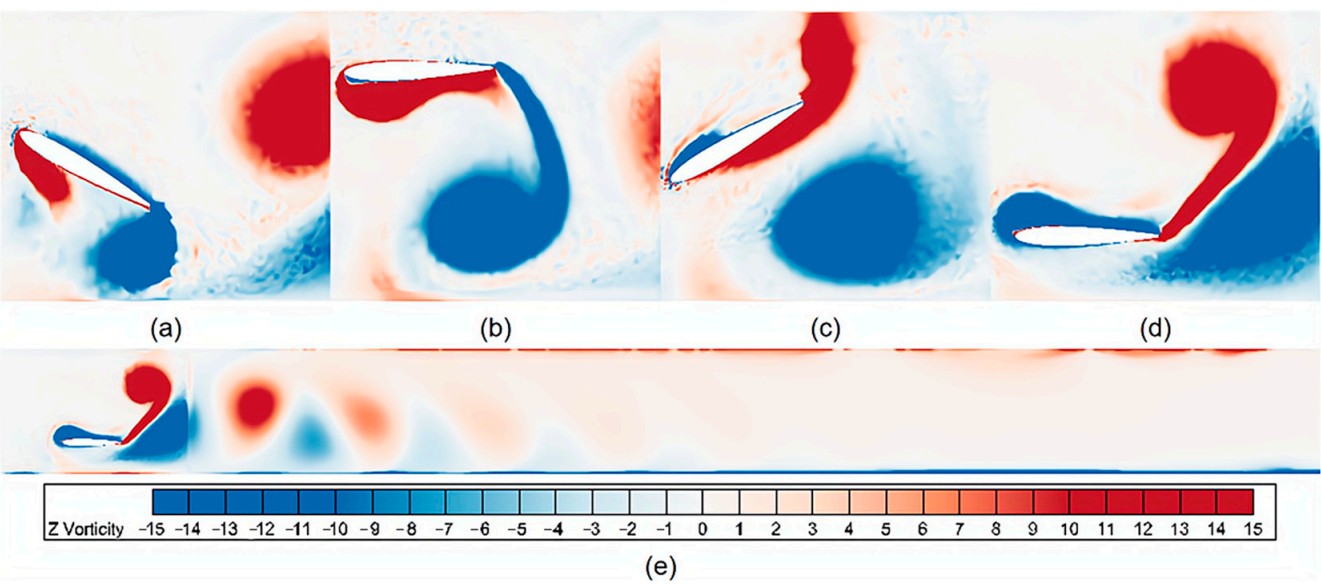

**Figure 13.** Velocity nephogram of the flow field during a single cycle at steady state (**a**) 1/4 T, (**b**) 1/2 T, (**c**) 3/4 T, (**d**) 1 T, (**e**) complete vorticity nephograms.

Figure 14 shows the velocity cloud of discrete-phase particles derived from fluent calculations at intervals of 1000 iterative steps, where the iterative step size is 0.01 s and the overall computation time is about 4.5 h. The picture depicts the distribution of discrete-phase particles over time for 60 s after particle injection, and the color distribution is determined according to the ensemble velocity of the particles when they are tracking-non-constant. In observing and analyzing the particle motion patterns, it is evident that, as particles continue to be injected, some clusters of particles with lower concentrations slowly discharge outward along specific velocity layers within the flow field, showing high sensitivity to velocity stratification. This phenomenon is attributed to the dissipative effect of vortex streets in the flow field, which generates thrust along the vortex center, particularly significant in regions with higher levels of vortex dissipation at the tail of the Kármán vortex street. Consequently, particles reach an equilibrium threshold in the Y-direction, where the gravity they experience balances out the thrust generated by vortex street dissipation.

Moreover, it is observed that some clusters of particles with higher concentrations slowly descend under gravity and exhibit fluctuating drifting along the direction of the flow velocity. This result can be attributed to the sinusoidal velocity curve generated by the flapping hydrofoil leading to velocity fluctuations in the flow field. Particles experience lateral drifting along the flow field due to the influence of flow velocity fluctuations. When the particle concentration is high enough, gravity overcomes the maximum thrust generated by the vortex street dissipation, causing particles to accelerate their descent vertically based on the magnitude of the thrust they experience. These denser particle clusters settle at the bottom and are slowly discharged outward along the flow velocity direction.

Particle clusters with an insufficient concentration or dissolved particles experience irregular settling drift due to velocity field fluctuations and turbulent flow. They are lifted by vortex effects and eventually settle back to the bottom or are directly discharged from the raceway. Comparatively, particle clusters with moderate concentrations are discharged at a faster rate, followed by those with lower concentrations. Higher-concentration particle clusters require overcoming bottom friction, resulting in slower discharge rates. Based

on the results and analyses presented above, the use of the flapping hydrofoil pumping model in raceway aquaculture significantly improves the drainage of suspended particles. Additionally, the formation of a downward-dissipating reversed Kármán vortex street in raceway aquaculture can effectively solve the problem of insufficient flow velocity caused by a large bottom shear stress. The thrust generated by the vortex dissipation can also aid in improving the deposition of suspended particles and increase the drainage capacity of the raceway aquaculture.

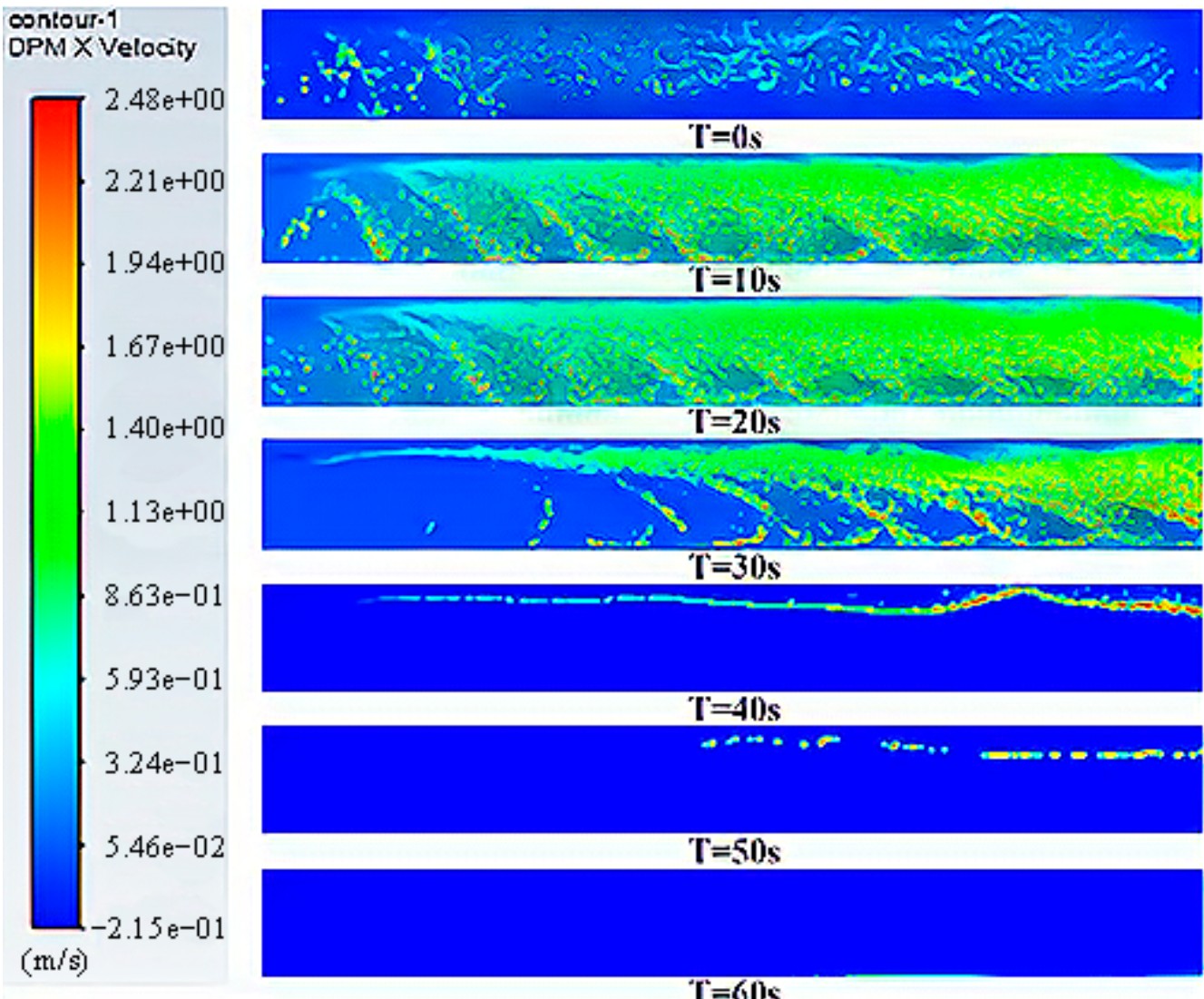

**Figure 14.** Schematic diagram of the effect of flapping hydrofoil pumping water on particle distribution.

*4.2. Impact of Pumping Depth on the Efficiency of Discharging Suspended Particles*

4.2.1. Variable Setting for Pumping Depth

This study examined the effect of the pumping depth on suspended particles. To avoid interference from other variables, the pumping frequency of the flapping hydrofoil was set at a constant 1.0 Hz. The total depth of the flow channel selected for this paper was 2 m, based on the actual situation of aquaculture raceways. To evaluate the efficacy of particles discharged from different pumping depths, the pumping depth ($h_p$) (Figure 15) was increased from 0.8 H to 1.2 H with 0.1 H intervals.

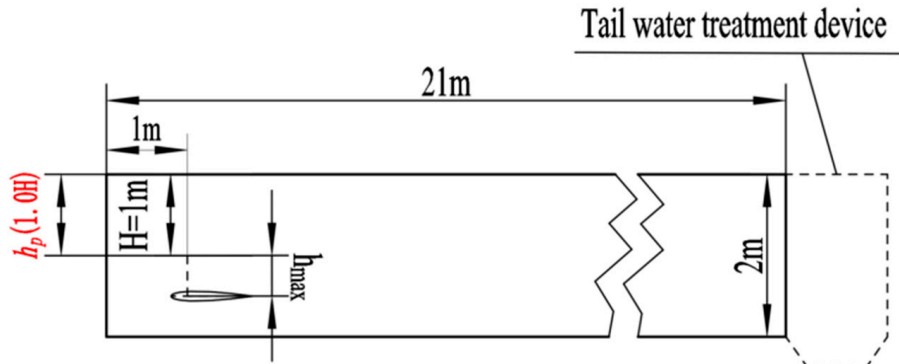

**Figure 15.** Schematic of pumping depth.

### 4.2.2. Effect of Pumping Depth on Particle Escape Rates and Escape Rates

Figure 16 shows how the escape rate varies with different pumping depths, indicating a significant impact of the pumping depth on the escape rate. As fluid flows due to the flapping hydrofoil, the particle escape rate ($E_R$) continuously increases until it reaches a maximum value, which is referred to as the maximum escape rate in this study. The maximum escape rate of particles gradually decreases with an increasing pumping depth. Of the chosen depth parameters, the particle discharge efficiency is highest at $h_p = 0.8$ H, where the maximum escape rate is 98.86% after 60 s of fluid participation in the flow. Conversely, the particle discharge efficiency is lowest at $h_p = 1.1$ H, with a maximum escape rate of 95.40%. It should be noted that the particle discharge effect of the flapping hydrofoil pump is superior at $h_p = 1.2$ H compared to $h_p = 1.1$ H, which deviates from the expected trend of the impact of the pumping depth on the particle discharge efficiency.

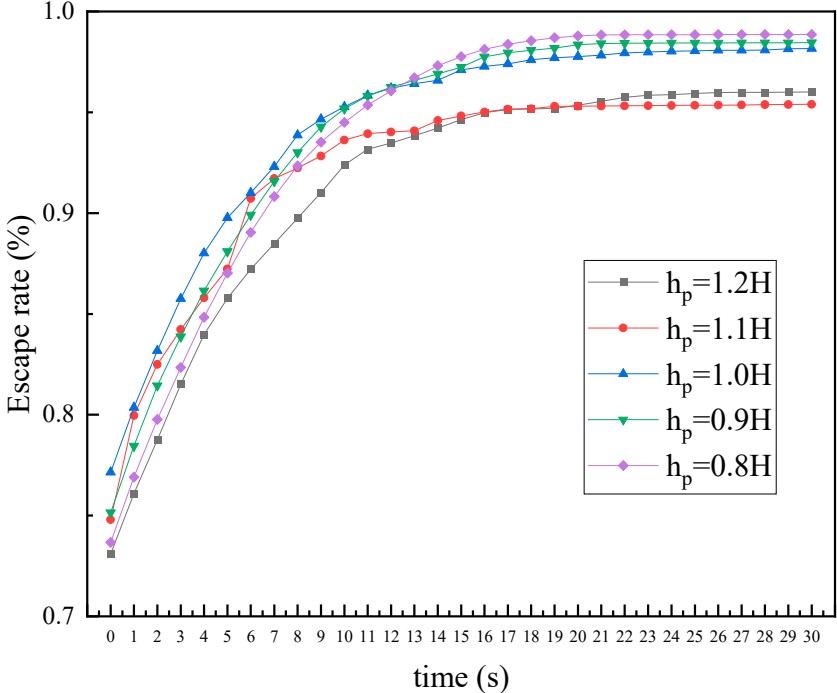

**Figure 16.** Effect of different pumping depths on the variation of escape rates.

Analyzing the above results, as shown in Figure 17, the maximum distance d of the reversed Kármán vortex street generated by the flapping hydrofoil motion gradually decreases as the pumping depth decreases at $h_p < 1.0$ H, making the initial position of the stable flow region generated by vortex dissipation more forward. At this time, the particles are affected earlier by the thrust generated during vortex dissipation, which avoids

premature deposition of the particles. However, at the same time, if the pumping depth was too shallow, the reversed Kármán vortex street would fall due to gravity, resulting in more vortex dissipation, which in turn resulted in insufficient thrust generated by the flapping hydrofoil, seriously affecting the particle discharge rate. Therefore, in the later study of the combined effect of the depth and frequency, too shallow a pumping depth significantly reduced the discharge efficiency at low frequencies. When $h_p > 1.0$ H, the increase in pumping depth exacerbates the flapping hydrofoil motion, which is disturbed by the wall shear at the bottom of the flow channel, thus increasing the instability of the vortex. The vortices generated by the flapping hydrofoil are squeezed by the bottom, resulting in a tendency for the reversed Kármán vortex street to rise and then sink. This results in increased uncertainty in the flow field and the premature deposition of some particles on the bottom surface due to turbulence in the flow field. Although the extent of the turbulent region at this point is larger than that at $h_p \leq 1.0$ H, the uneven arrangement of the vortex streets reduces the overall flow velocity and results in a small number of particles being prematurely deposited on the bottom of the flow channel, reducing the efficiency of the particles being discharged.

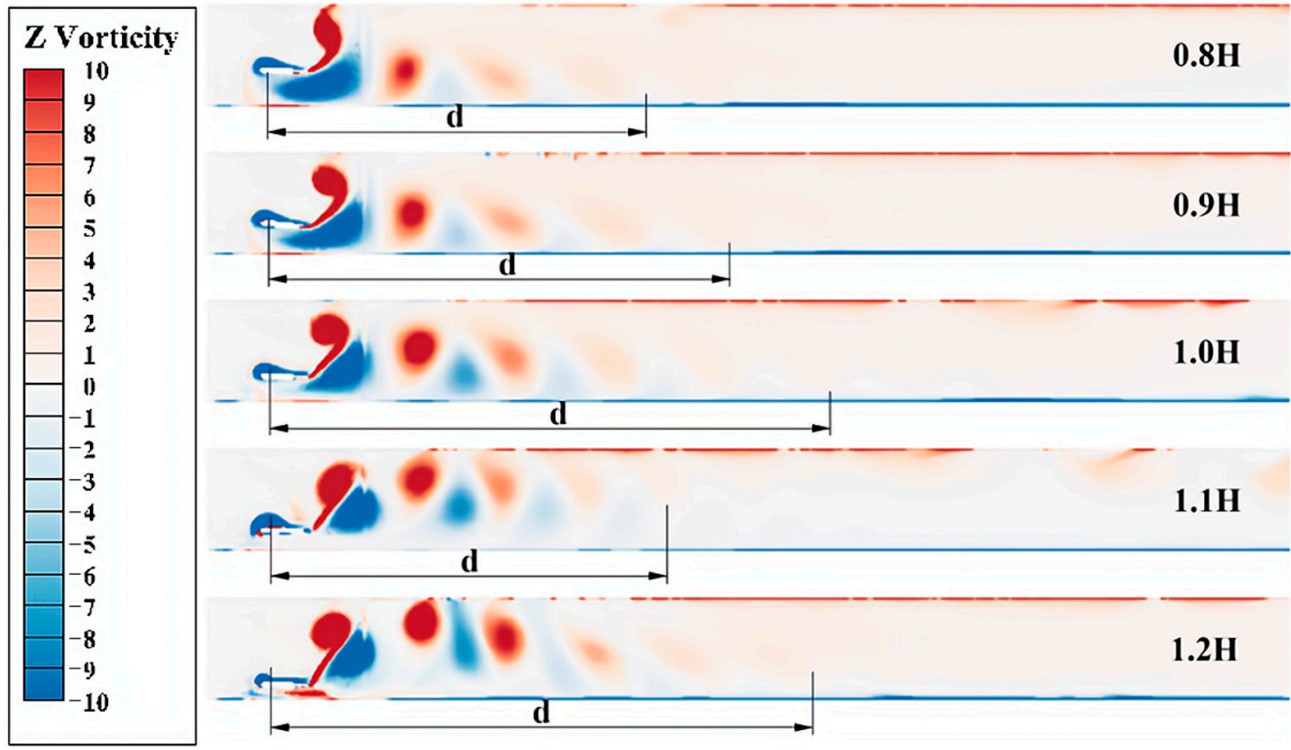

**Figure 17.** Nephogram of runner vorticity at different pumping depths for 1.0 Hz pumping frequency.

Figure 18 shows the velocity profiles of the runners for different pumping depths at a pumping frequency of 1.0 Hz When comparing the vortex cloud and velocity cloud diagrams at $d_p = 1.1$ H in Figures 17 and 18, it can be observed that the flapping hydrofoil motion is less affected by the bottom compared to 1.2 H. The vortex generated during the motion is relatively complete. However, when the average velocity is stable, the near-bottom maximum flow velocity is not enough to create obvious flow stratification, resulting in a slight decrease in total particulate emissions. Comparatively, when $d_p = 1.2$ H, the Kármán vortex street structure is disrupted by significant wall effects [26], caused by the bottom wall, on the flapping hydrofoil motion. However, this disruption leads to a more pronounced stratification of the flow velocity in the downstream section of the channel, which increases the particle discharge efficiency.

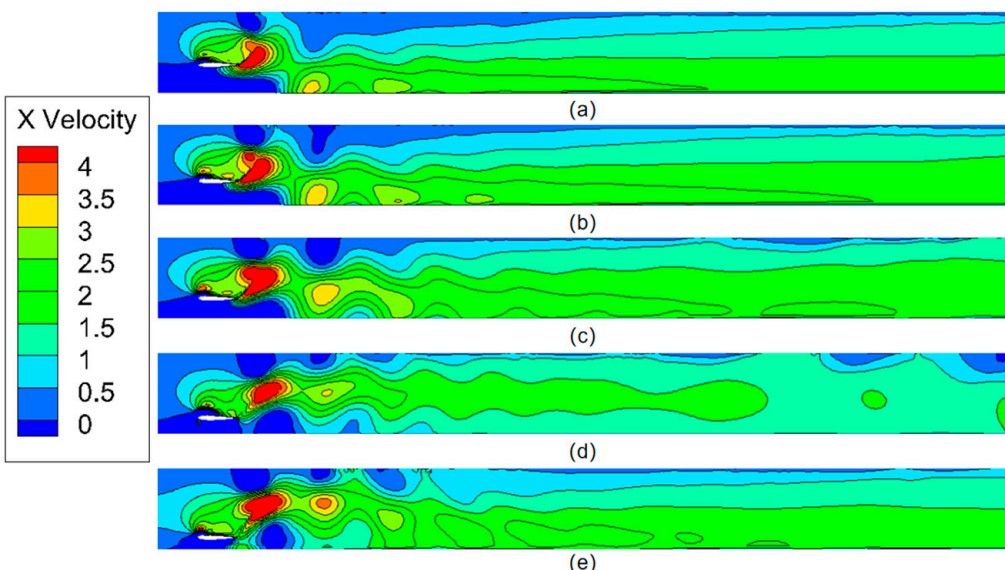

**Figure 18.** Velocity nephograms of runners at different pumping depths for 1.0 Hz pumping frequency; (**a**) $d_p$ = 0.8 H, (**b**) $d_p$ = 0.9 H, (**c**) $d_p$ = 1.0 H, (**d**) $d_p$ = 1.1 H, (**e**) $d_p$ = 1.2 H.

In order to more intuitively analyze the effect of the flapping hydrofoil on the effect of the effluent, it is necessary to correlate it with the discharge velocity, taking into account that the escape rate of the particles does not directly reflect the effect of the flapping hydrofoil on the discharge velocity; therefore, this paper defines the growth rate of the escape rate over time as the escape velocity rate. Figure 19 shows the variation in the escape velocity rate at different pumping depths. It can be observed that, at a pumping depth of 1.1 H, the discharge rate curve is significantly better than that of other working conditions, and the escape rate reaches the peak value earlier. At this depth, the reversed Kármán vortex street generated by the flapping hydrofoil is more complete and extends further before dissipating due to the compression of the bottom wall.

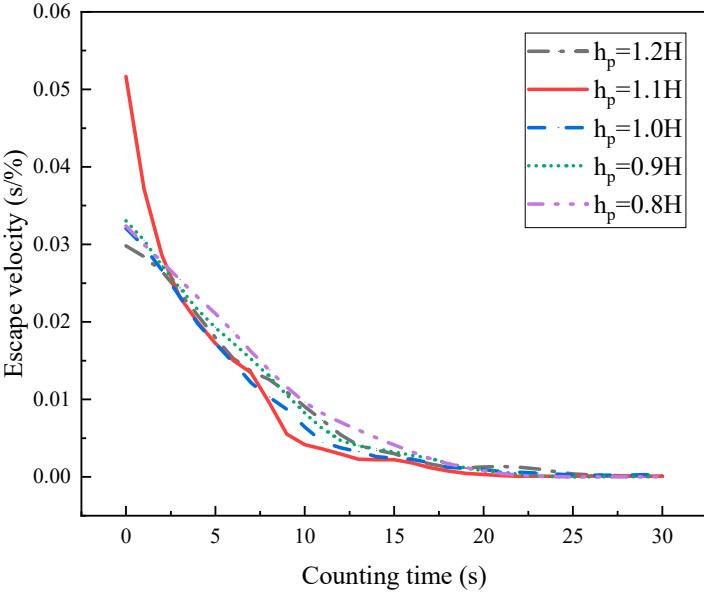

**Figure 19.** Effect of different pumping depths on the variation in escape velocity rates.

This study indicates that the optimal pumping depth should not be too deep or too shallow. If the pumping depth is too shallow, the flapping hydrofoil will not generate enough thrust. Conversely, if the pumping depth is too deep, the flapping hydrofoil

movement will be disrupted by the bottom surface, leading to an increase in the particle deposition rate. To meet specific discharge requirements, different flapping hydrofoil pumping depths can be selected. To achieve the rapid discharge of suspended particulate matter, it is recommended to select $d_p$ = 1.1 H. This facilitates quick particle discharge in a short time. Conversely, if a higher total final particulate discharge is required, it is recommended to select $d_p$ = 0.8 H. This results in particles being discharged at a relatively slow velocity rate, but a higher maximum escape rate will be achieved.

### 4.3. Analysis of the Combined Effect of Pumping Depth and Pumping Frequency

Simulation experiments were conducted to study the joint effect of the flapping hydrofoil pumping depth and pumping frequency on the particle discharge efficiency. The experiments were carried out for different pumping frequencies at each $d_p$, ranging from 0.8 H to 1.2 H (with intervals of 0.1 H). The frequency was varied from 0.5 Hz to 2 Hz (at intervals of 0.25 Hz) and then from 2 Hz to 3 Hz (at intervals of 0.5 Hz). To facilitate this study, this paper defines low frequency as 0.5 Hz–1.25 Hz, medium frequency as 1.25 Hz–2 Hz, and high frequency as above 2 Hz due to the limited frequency range of flapping hydrofoil. This definition ensures clarity in the research.

When comparing the escape rate versus frequency curves at different pumping depths in Figure 20, it is evident that the discharge efficiency of particles increases with an increase in the pumping frequency. However, the improvement in discharge efficiency slows down after 2.0 Hz–3.0 Hz. There is a joint effect between the pumping depth and frequency. At a low-frequency condition of 0.5 Hz, the particle discharging effect is better when the pumping depth is closer to the middle position (1.0 H). However, when the depth is 1.2 H, the flapping hydrofoil motion is disturbed by the wall surface, resulting in obvious fluctuations in the escape rate with an increase in frequency, particularly between 1.0 Hz and 1.5 Hz. The pumping effect of flapping hydrofoils at a depth of 1.2 H improved significantly with increasing frequency and was optimal at a high frequency of 3.0 Hz.

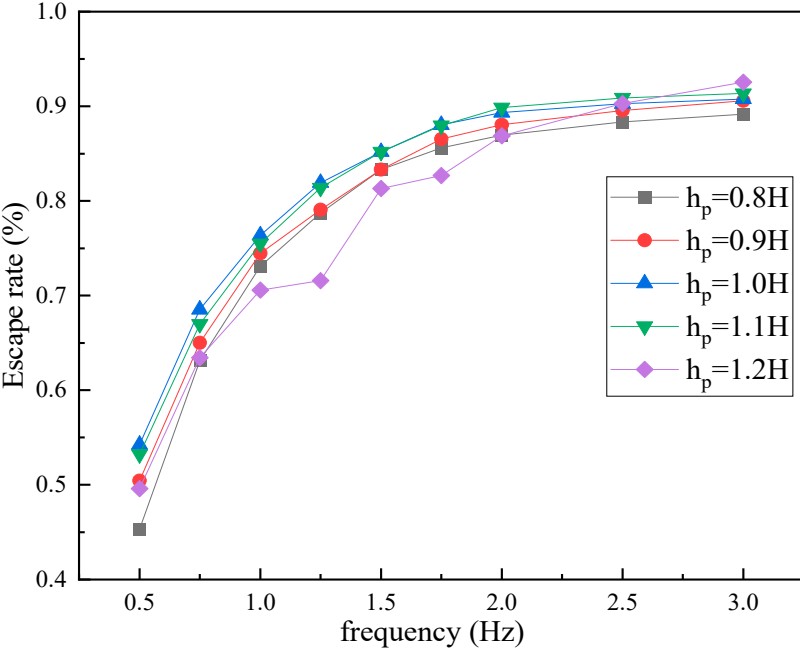

**Figure 20.** Curves of escape rate versus frequency at 30 s of particle placement at different pumping depths.

Of the five groups of data with varying pumping depths, the two groups at pumping depths of 1.0 H and 1.1 H exhibited greater stability with frequency enhancement and did not produce significant fluctuations with increasing frequency. The 1.0 H pumping depth performed better at lower frequencies (0.5 Hz–1.25 Hz). This study found that, as the

frequency increased from 1.25 Hz to 5 Hz, the particle escape rate at $d_p$ = 1.1 H exceeded that at 1.0 H, reaching a maximum value of 91.8% at 3 Hz, which is 1.1% higher than the particle escape rate at 3 Hz for $d_p$ = 1.0 H. At a frequency of 2.0 Hz, the particle escape rates for all four pumping depths were close to the stable value, except for the pumping depth of 1.2 H

Upon analysis of the results, it was found that the flapping hydrofoil motion generates a well-arranged reversed Kármán vortex street when the pumping depth is less than 1.1 H. The structure of the vortex street generated by the flapping hydrofoil remains relatively constant as the frequency increases; however, the size of the vortex volume does increase with the frequency. The increase in the vortex volume enhances the thrust generated by the vortex during dissipation, resulting in improved particle effluent discharge. The vorticity cloud diagrams of the flow field at different pumping frequencies at a 1.2 H pumping depth in Figure 21 show that an excessive flapping hydrofoil depth disturbs the flapping hydrofoil's motion due to the violent wall effect at the bottom. At this depth, the flapping hydrofoil produces two pairs of interacting leading-edge vortices and trailing-edge vortices in two motion cycles. The leading-edge vortices merge pair by pair to form a larger counter-rotating vortex during motion. However, the trailing-edge vortices experience severe dissipation due to the bottom surface interference. When the frequency reaches 2.0 Hz–3.0 Hz, the increase in the flow velocity is directly proportional to the frequency. However, the vortex shape remains unaffected by the increase in the frequency, resulting in an insignificant increase in the effluent effect. Additionally, the increase in frequency causes the particles to reach the maximum escape rate earlier due to the increase in the escape velocity rate, which also affects the analysis. Examining the velocity cloud diagrams in the x-direction at different pumping frequencies and a pumping depth of 1.2 H, shown in Figure 22, it is evident that the velocity field structure in the runway remains relatively constant with an increasing frequency. However, the flow velocity stratification at the end of the runway becomes more pronounced, which is a significant factor contributing to the increase in the particle escape velocity rate.

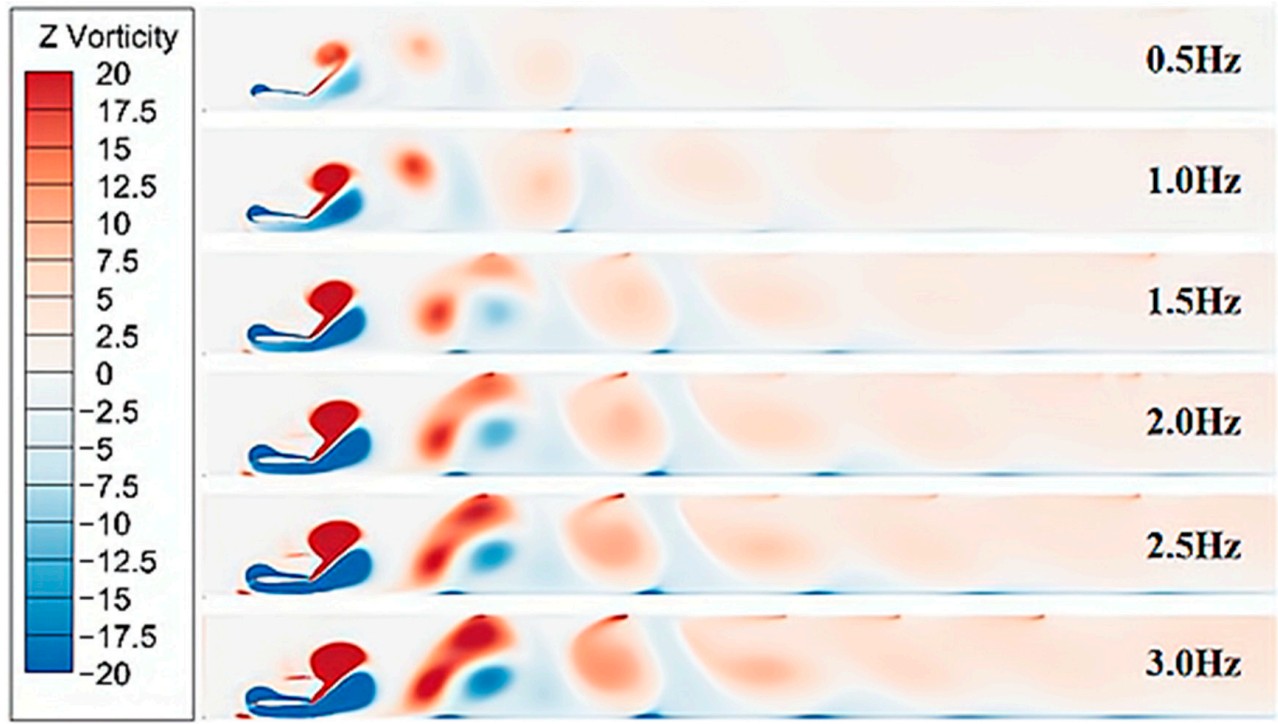

**Figure 21.** Vorticity cloud diagrams of the flow field at different pumping frequencies for the pumping depth of 1.2 H.

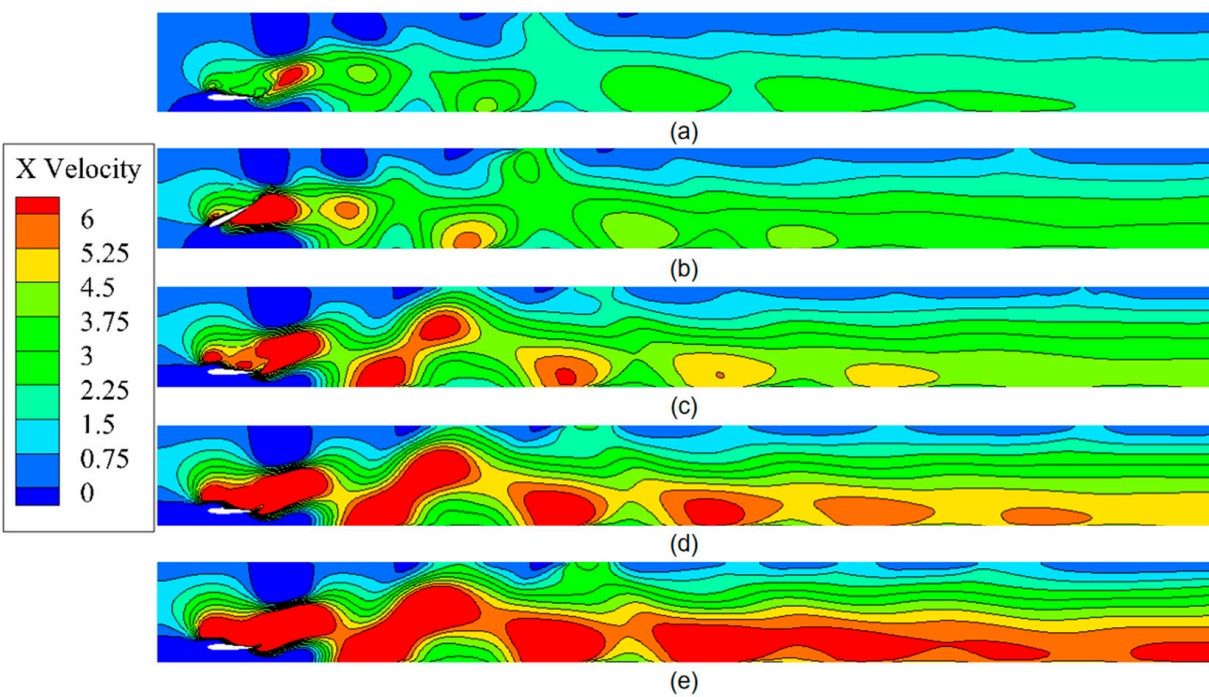

**Figure 22.** Velocity cloud diagrams of the flow field at different pumping frequencies for the pumping depth of 1.2 H. (**a**) 1.0 Hz, (**b**) 1.5 Hz, (**c**) 2.0 Hz, (**d**) 2.5 Hz, (**e**) 3.0 Hz.

The discussion of the above results leads to the conclusion that higher flapping frequencies result in a higher efficiency of discharging suspended particles. Additionally, there is a combined effect of the flapping hydrofoil's depth and frequency on the discharge of suspended particles. At lower frequencies (0.5 Hz), the closer the flapping hydrofoil depth is to 1.0 H, the better the discharge effect on suspended particles. As the frequency increases, deeper flapping hydrofoil depths result in a better discharge efficiency of suspended particles. However, excessively high pumping frequencies inevitably lead to increased instability of the mechanism and the wastage of energy. Furthermore, at higher pumping depths, the discharge performance of the flapping hydrofoil at high frequencies is superior. However, at this point, due to bottom wall interference, the structure of the Kármán vortex street is disrupted, leading to a decrease in pumping efficiency. This paper recommends a combination of a pumping depth of 1.1 H and a pumping frequency of 2.0 Hz, as it achieves a high efficiency in raceway discharge while maintaining flow field stability.

## 5. Conclusions

This study examines the application of a flapping hydrofoil in raceway aquaculture through numerical simulation using CFD. It investigates the influence of different parameters of the flapping hydrofoil on the settling and drifting characteristics of suspended solids in water. Through simulation experiments with various depths and frequencies, the following conclusions are drawn:

- The flapping hydrofoil generates a gradually dissipating Kármán vortex street during the pumping process, effectively addressing the issue of insufficient flow velocity in ecological water bodies. Additionally, the thrust generated by vortex dissipation enhances the discharge efficiency of suspended particles;
- Excessive depth of the flapping hydrofoil pumping leads to significant interference from the bottom, while a shallow pumping depth results in the downward dissipation of the Kármán vortex street due to gravity, consuming more energy and resulting in inadequate thrust generated by the flapping hydrofoil;

- At a pumping depth of 1.1 H, the flapping hydrofoil motion produces a more complete and stable reversed Kármán vortex street, which effectively reduces the effect of runway end-wall shear on the near-bottom flow velocity and enhances the overall flow velocity of the raceway, which in turn leads to a higher rate of escape velocity of suspended particles at this point;
- The depth and frequency of the flapping hydrofoil pumping water have a combined effect on the particle discharge. At low frequency (0.5 Hz), the closer the pumping depth of the flapping hydrofoil is to 1.0 H, the better the effect of the flapping hydrofoil discharge; as the frequency increases, the deeper the pumping depth of the flapping hydrofoil is, the better the effect of the suspended particles' discharge;
- In raceway aquaculture sewage discharge, this paper suggests the reference values of pumping depth and frequency of 1.1 H and 2.0 Hz. This combination of pumping parameters for flapping hydrofoils can produce stable reversed Kármán vortex street structures, and without energy wastage due to a high frequency without significant pumping efficiency improvement, the flapping hydrofoils can obtain good pumping efficiency and suspended pollutant discharge results.

**Author Contributions:** E.H. presents the main directions of the influence of the pumping depth and frequency parameters of flapping hydrofoils on the distribution of suspended particles in water; M.X. was involved in the full writing of the manuscript as well as in data processing; T.W. provided ideas for detailing as well as simulation design, and participated in model construction. Y.S. contributed to the model construction and simulation experiments and validation. C.L. provided writing ideas; Q.S. reviewed the manuscript. All authors have read and agreed to the published version of the manuscript.

**Funding:** This research was funded by the National Natural Science Foundation of China (Grant No. 51976202, 61772469) and the Zhejiang Provincial Key Research and Development Project (Grant No. 2021C03019).

**Data Availability Statement:** All data, models, and code generated or used during the study appear in the submitted article.

**Acknowledgments:** The authors gratefully thank the National Natural Science Foundation of China and Zhejiang Provincial Key Research and Development Project for their financial support.

**Conflicts of Interest:** The authors declare no conflicts of interest.

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
