# Peer review of "Investigation of the Effect of Pumping Depth and Frequency of Flapping Hydrofoil on Suspended Matter Discharge Characteristics"

_machines, doi:10.3390/machines12050300_

Round 1
Reviewer 1 Report
Comments and Suggestions for Authors
The manuscript is scientifically sound and has merit.
The are various shortcomings currently that need to be addressed. The paper suffers from many editorial shortcomings, cf:
1. '...Research [5] utilizing DPIV visualization of flow' in line 37. Who is Research?
2. '...Regarding the flapping hydrofoil device, Knooller. R[14]found that the angle of...' who is .R?
3. 'Du et al. [16]conducted' space required after [16].
4. Fig 2 requires upgrade - seems cut and paste from elsewhere.
5. Eqs 1-4 seem cut and pasted from a different source. Matching of style of the manuscript required.
6. Fig.3 similar to Fig.4
7. Eq 8 is completely out of alignment
8. line 246 starts with upper case.
There are many other examples that require attention and distract the reader from assessing the scientific substance of the manuscript. The paper elevates itself when it comes to the scientific discussion of the the content.
Here:
1. There is no in depth of the usage of the CFD. Grid meshing is provided but computational times and convergence of the calculations must be exposed. Currently there is only discussion of the results.
2. The calculations are seemingly 2 dimensional. If they are 3D calculations this should be explicitly stated within the manuscript. The computational times and the convergence criteria should be explicitly mentioned too.
3. The deviations observed in Figs 2-3 should be addressed. The presented results seem to agree well but there is a distinction. The authors should address this point further.
4. I liked the description presented in subsection 4.1 but is it 2D or 3D? If 3D the meshing adopted is low.
5. Maybe the authors should provide a brief description of what nephogram is within the context of their work.
6. Figure 14 should. benefit from a better quality screen shot.
7. Figure 14: how were the transient results presented here obtained? Iterations? Time required? Hardware upon which the calculations were performed? All these questions require answers.
8. Line 432:
'This paper defines the growth rate of the escape rate with time as the escape velocity rate.'
The authors should elaborate on this. I mean what is the reason behind this definition? Why is a definition needed here?
9. Bullet points 3 and 5 of the conclusions: The authors should add a sentence on each bullet point to substantiate the assertions made here.
10. Is the software proprietary? This should be stated explicitly and more detail should be provided in section 3.
The manuscript uses experimental data to validate the simulative reults. This is good but the description presented is very brief.
The manuscript needs to be improved in order to proceed to publication. In my opinion it should not be published in its. present form.
Author Response
Dear reviewers.
Thank you for your acknowledgement of our paper and your valuable comments and suggestions. We have carefully considered your feedback and have revised and improved the paper accordingly. Below are the specific revisions and responses we have made in response to your comments:
In terms of editing, based on your comments, we have made the following revisions to the thesis:
- In line 37, the name of the researcher is not specified, it should be "Anderson, J. M.", which has been revised.
- The name of the researcher here is R Knoller, which was incorrect in the previous manuscript and has been corrected.
- the formatting problems here have been corrected, and the full text has been checked for similarities in content and corrected where the same problems existed.
- Figure 2 has been revised and adjusted.
- Formatting inconsistencies previously existed, and the formatting has been adjusted for consistency.
- Regarding the similarity between Figure 4 and Figure 5, some adjustments have been made to Figure 4 to increase the differentiation between the two and their respective highlights.
- The position of the formula has been adjusted to make it consistent with the format of other formulas.
- Previously, the first letter in line 246 (line 211 after the revision) was not capitalised, which has been corrected. In addition initial capitalisation throughout the article was checked and corrected.
As for the content of the article, we have discussed and analysed your comments and have made the following changes based on them:
- In the original paper, there is insufficient discussion on issues such as computational time and computational convergence. Based on this issue, we have included an introduction on the computational time required for CFD calculations and computational convergence in the paper. Information on the computational resources used for the simulation (e.g., computer configuration) and the time required to run the simulation is also provided. In addition, we discuss the convergence of the simulations under different parameter settings, including the convergence criteria and the number of iterations required for convergence.
- With regard to computational issues, we have not previously made it clear in the paper that we use 2D computation for all our simulations due to the fact that 3D computation takes up a lot of exponentially growing computational resources and does not have an impact on the results. We have now added a clarification question about the dimensionality of the computations in subsection 3.2, which we hope will make our computational dimensions clearer.
- Regarding the issue of bias in Figure 10, in conjunction with the reviewer's suggestion, the previous manuscript did not clarify the reasonableness of the bias in the results, and in the revised manuscript, we explain why the bias here has little impact on our results (i.e., our study is more concerned with the consistency of the motion trends of the particles than with the accuracy of the particle calculations).
- In subsection 4.1 we also include a discussion of model dimensionality, which admittedly requires the reader to first specify whether the model is two- or three-dimensional, however the previous manuscript clearly left more to be desired in this section.
- "Nephogram" is the concept of the overall picture of the flow field in CFD post-processing, perhaps we can replace it with cloud diagram, but after discussion, we prefer to use "nephogram".
- The quality of the screenshot in Figure 14 is low due to the format of the image exported in the previous work. We recalculated this part of the simulation, remade Figure 14, and used AI technology to increase its clarity. However, we were unable to further optimise the clarity of the image because the original image clarity itself was not high. In addition, some of the visual blurring of the picture may be due to the dispersion of particle clusters, so the lack of picture clarity here is unavoidable.
- Regarding the transient results in Fig. 14, our previous manuscript did not explain how the images were obtained, and in the revised manuscript we explained how the images were obtained, and described the iteration time steps and the iteration intervals for the acquisition of the images, in addition, the hardware for the computation was already mentioned in the previous revisions, and so we did not add the description of the hardware here.
- The reason for the definition of escape velocity rate was not sufficiently explained in the previous revision, so the reason for the definition was added in the revised version.
- For the third and fifth points in the conclusion, the reviewer pointed out that we had insufficient arguments. We have revised and refined them by adding explanations of the reasons behind the conclusions.
- In section 3, we provide the licence of the software we used (ansys-fluent), which we obtained before using it, and which is commercial rather than proprietary, so we are not able to provide specific details about the software in the paper, which are freely available on the official ansys website.
Thank you for your review and feedback on our paper, we value your comments and have made every effort to make changes. If you have any further suggestions or comments, we would be happy to hear them and make further improvements. We look forward to hearing from you.
Sincerely,
Mingwang Xiang
Reviewer 2 Report
Comments and Suggestions for Authors
The article provides a multi-variant analysis of a practical engineering problem.
The conclusions from the conducted analysis may be useful for people designing and operating similar systems.
However, the adopted model and the analysis performed have certain shortcomings, which are listed below.
Fig. 1 is incomprehensible
How Re is defined (line 190)? At least two Re numbers can be defined in the model: one associated with the hydrofoil and the other associated with the solid particle (grain).
Formula 6 is obtained assuming that the solid body has the shape of a sphere. Meanwhile, real contaminants may have shapes very different from the shape of a sphere.
The model assumes (line 192) that the grain diameter is constant and is dp = 0.025 mm. Meanwhile, real impurities (grains) have different diameters (which can be described as a grain size distribution). Additionally, it should be noted that fish faeces can be break down into smaller particles, so their diameters will decrease over time.
The model assumed (line 195) a constant value of the drag coefficient (c=0.47). This coefficient depends on the grain roughness and also on the Re number (which is not constant - see notes above).
Pattern 8: suggestion: adopt some symbols and use the symbols in the formula.
Line 193 assumed a density of 1100 kg/m3. Line 290 talks about quartz sand, the density of which is much higher.
Complete: how the x y and z axes were assumed?
It would be very valuable to know how the analyzed pump/tank operating parameters affect on energy consumption.
Author Response
Dear reviewers.
Thank you for your acknowledgement of our paper and your valuable comments and suggestions. We have carefully considered your feedback and have revised and improved the paper accordingly. Below are the specific revisions and responses we have made in response to your comments:
- Figure 1 may have been inconvenient for readers to understand in previous manuscripts due to unclear lines or insufficient descriptions, we have improved the picture and made a note of the picture in the text.
- Previously, we made a mistake in the definition of Reynolds number in this part, we should determine the corresponding settling velocity formula according to the Reynolds number of the particles, but the fluttering Reynolds number was used in the previous manuscript, which is a mistake, we have made a modification: the way of defining the Reynolds number of the particles is written, and the corresponding settling velocity formula is re-determined according to the Reynolds number of the particles, which is corrected in the subsequent calculations, and we get the The corrected particle mass flow rate Q. In the simulation, this part of the calculated value is not directly involved in the calculation process, so the change of the mass flow rate and settling velocity values here will not affect the results of our later study.
- In the calculations, we take spherical suspended particles as an example, but in reality, different shapes can lead to discrepancies between our calculations and the actual values. We have found that elliptical particles, for example, are shifted to a different extent when moving compared to spherical particles, which can have a significant impact on our results. According to the previous literature review, we found that the shape of small-sized particles has a greater effect on the velocity when they rub against the wall, but in this paper, if the particles are in contact with the bottom surface, the particles will reach the state of approximate stopping of the motion in the form of deposition, and the difference in the shape of the particles after contacting the bottom surface in this case is almost impossible to affect the results of our study. And the deviation due to the different shapes of the particles when they move in the medium is obviously within the controllable range and will not significantly affect the trend of the particle escape rate in our study. Therefore, for the convenience of our study, we have simplified all the particles to spherical particles. This we have added in the revised manuscript the previously missing description.
- The previous manuscript did not make sense in terms of particle size, so we have modified the particle size setting in this section to describe it as a size distribution, and have re-explained the issue of particle size reduction and dissolution of particles.
- The previously existing problem of resistance coefficient as a constant value (c=0.47), due to the change of Reynolds number, we adopted another formula, and in the new formula, c is not involved in the calculation. In addition, the reason for previously setting c as 0.47 is to refer to the empirical formula in the literature [26], which mentioned that the drag coefficient changes with the change of Reynolds number, but the change is small in the selected range of Reynolds number, ranging from 0.44 to 0.5, so the average value of 0.47 is taken.
- According to your comments, we renamed the variables in Eq. 8 (Eq. 9 in the revised version) as some abbreviated symbols, and explained the symbols when mentioning these variables later.
- You pointed out that "Line 193 assumed a density of 1100 kg/m3. Line 290 talks about quartz sand, the density of which is much higher. This is due to the fact that our simulations of quartz sand were done only to ensure the reliability of our discrete phase simulations. As the density increases, it is obvious that the settling velocity of the particles will become larger, and thus the calculation error will also become larger, so our original idea is to ensure the calculation accuracy of the particle phase at high density, although there is a certain deviation in our simulation results, the trend is still consistent in the case of simulation of high-density particles, and the numerical error is also kept within 0.0316 m/s. For this, we have explained and improved the simulation calculation of quartz sand more comprehensively in the chapter (3.4) of method validation.
- According to your question "how the x y and z axes were assumed?", we found that the selection of x,y and z axes was not explained before, and we added the coordinate descriptions in the schematic diagram of the model, which makes the readers clear about the selection of our coordinates.
Finally, thank you again for reviewing and providing feedback on our paper and for recognising the content of our research, we take your comments very seriously and have made every effort to revise them. If you have any further suggestions or comments, we will be more than happy to hear them and make further improvements. We look forward to your reply.
Sincerely,
Mingwang Xiang
Reviewer 3 Report
Comments and Suggestions for Authors
This paper utilizes the CFD numerical simulation to investigate the effect of different pumping depths and pumping frequencies of the flapping hydrofoil device on the suspended solids in the water body. In overall, this paper is well structured and meaningful in optimization design of flapping hydrofoil devices in raceway aquaculture systems. However, there are still some issues that should be fully addressed.
1. The article should undergo grammar revision.
2. The Realizable k-epsilon model is selected to simulate the turbulent flow, could the author explain the reason for choosing this model?
3. Please provide the relative error percent between theory and experiments in Figure 10. In particular, I suggest that the authors explain if the assumptions involved in the model are (or are not) justified in the experimental setup.
4. The resolution of Figure 14 is not sufficient.
5. The abstract section needs to add the practical engineering significance of the research work.
6. It is necessary to explain the physical meaning of Kármán vortex streets and their correlation with the efficiency of propeller propulsion.
7. In the paper, it is necessary to explain how to conduct research on Kármán vortex streets and how to display them.
8. Figure 1 is not cited in the introduction.
9. In the last paragraph of the introduction, it is generally necessary to explain the experimental methods, such as what experimental methods and simulation methods (turbulence models, etc.). This also includes the means of verification, as it is a reliable basis for simulation results.
10. The RNG k—ε turbulence model is mentioned in reference [23], but the literature is a bit outdated. Suggest adding the latest literature( https://doi.org/10.1016/j.cep.2024.109775 ).
Comments on the Quality of English Language
NO.
Author Response
Dear reviewers.
Thank you for your acknowledgement of our paper and your valuable comments and suggestions. We have carefully considered your feedback and have revised and improved the paper accordingly. Below are the specific revisions and responses we have made in response to your comments:
- You pointed out that our previous manuscript had a grammatical deficiency, and we have rechecked and revised the grammar of the whole text, which hopefully will make our article more fluent to read.
- In the revised manuscript, when we mentioned the Realizable k-ε model, we explained in more detail why we adopted this model and how it compares with other models.
- We add the maximum error value between theory and experiment in Figure 10. and further explain the reasonableness of the assumptions involved in our model in the experimental setup.
- The screenshot in Figure 14 is of low quality due to the format of the image export in previous work. We re-calculated the simulation for this section, remade Fig. 14 and used AI techniques to increase its clarity. However, we were unable to further optimise the clarity of the image because the original image clarity itself was not high. In addition, part of the visual blurring of the picture may be due to the dispersion of particle clusters, and the lack of clarity in this part of the picture is unavoidable.
- You mentioned that the practical engineering significance of our study was not clearly stated in the previous abstract, according to your suggestion, we added the practical engineering significance of our study for runway aquaculture effluent and ecological water purification in the revised manuscript.
- Referring to your suggestion, we think it is necessary to explain the physical meaning of the Kalman vortex street and its correlation with the wing propulsion efficiency. In the revised version, when the Reversed Kármán Vortex Street is mentioned for the first time in subsection 4.1, we explain in detail its physical significance and its relevance to the wing propulsion efficiency mentioned in this paper.
- You mentioned that "In the paper, it is necessary to explain how to conduct research on Kármán vortex streets and how to display them." In the revised draft, we explain the physical significance of the anti-Kármán vortex streets and their relevance to flap propulsion efficiency, along with how we research and display them. In the revised paper, we explain how to conduct research on Kármán Vortex Streets and how to display them, after explaining the physical significance of Reversed Kármán Vortex Streets and their relevance to the propulsion efficiency of flapping wings.
- You pointed out that "Figure 1 is not cited in the introduction." We have revised part of the introduction and cited Figure 1, and checked whether the citation of all the other pictures and their corresponding positions are correct.
- We fully agree with your suggestion of "In the last paragraph of the introduction, it is generally necessary to explain the experimental methods, such as what experimental methods and simulation methods (turbulence models, etc.). This also includes the means of verification, as it is a reliable basis for simulation results.", and for this reason we have completely revised the end of the introduction, explaining the computational models and validation methods we have used, as well as some other details of the study, which we hope will make our work more clearly reflected in the introduction.
- You pointed out that our reference [23] is outdated, and we would like to cite the reference you suggested, but we found that the turbulence model used in that paper is RNG k-ε, which is not in line with the computational model we used in our study, and for this reason we cite another paper that mentions a Realizable k-ε turbulence model.(https://doi.org/10.1007/s10652-018-9637-1)
Finally, thank you again for reviewing and providing feedback on our paper and for recognising the content of our research, we take your comments very seriously and have made every effort to revise them. If you have any further suggestions or comments, we will be more than happy to hear them and make further improvements. We look forward to your reply.
Sincerely,
Mingwang Xiang
Round 2
Reviewer 1 Report
Comments and Suggestions for Authors
I think that the authors have significantly improved the quality of their manuscript.
There might be some minor editorial editing required but I think that the manuscript can now proceed to publication.
Author Response
Dear Reviewer,
We hope this message finds you well. We would like to express our sincere gratitude for your positive feedback and approval for the publication of our manuscript titled "[Manuscript Title]." Your acknowledgment of the improvements made to the manuscript is truly encouraging and reinforces our dedication to delivering high-quality research.
We have taken note of your suggestion regarding minor editorial editing, and we will ensure that any necessary adjustments are made to enhance the overall clarity and readability of the manuscript.
Your support and constructive feedback throughout the review process have been invaluable to us, and we greatly appreciate the time and effort you have invested in evaluating our work.
Once again, thank you for your positive assessment and for your confidence in the readiness of our manuscript for publication. We look forward to contributing our research findings to the scientific community through your esteemed journal.
Best regards,
Mingwang Xiang
Reviewer 2 Report
Comments and Suggestions for Authors
The Reynolds number is defined using the kinematic (liquid) viscosity.
Kinematic viscosity can be calculated as the ratio of dynamic viscosity to density - but both of these values are properties of the liquid. The authors propose (formula 6) to calculate the kinematic viscosity using the dynamic viscosity of the liquid and the density of the solid (instead of the liquid). This definition is incorrect (and also has no physical interpretation).
The model assumes (ver 1: line 192, ver 2: line 204) that the grain diameter is constant and amounts to dp = 0.025 mm. In the review for ver1, I pointed out that real contaminants have different diameters (which can be described as a grain curve). Additionally, it should be noted that fish excrement tends to break down into smaller particles, so their diameters will decrease over time.
the authors did not respond to this comment
Grain speed (formula 7) depends on the grain roughness. I pointed this out in my review for version 1.
The authors did not respond to this comment.
Author Response
Dear Reviewer,
We hope this message finds you well. We are writing to provide our responses to the comments and suggestions raised during the second round of review of our manuscript titled "Investigation of the effect of pumping depth and frequency of flapping hydrofoil on suspended matter discharge characteristics" for which we are grateful for your continued engagement and valuable feedback.
- You're absolutely right that kinematic viscosity can be calculated as the ratio of dynamic viscosity to density – but both of these values are properties of the liquid. I made a mistake in equation 6, so I've gone back to the reference and double-checked. It looks like the density should be that of water, so I've updated the equation. Thanks so much for pointing that out!
- On the issue of model assumptions, our previous revision may not have been obvious enough (lines 176-184 of the new revised version). Previously we revised the mean diameter of particles to 0.5mm and, based on your suggestion, explained the range of particle variability and why we did not account for particle disintegration and fragmentation (this is because particle disintegration and fragmentation is a stochastic process that would affect the accuracy of our findings). We have attempted to include particle disintegration and fragmentation in our model, but this would have resulted in haphazard fluctuations in the escape rate of particles over time, proving that this is a stochastic process. However this stochastic process can lead to our findings not being applicable in all cases (which is why we have considered smaller values when averaging), and given the problematic nature of this part of the modification, we have included a more detailed description of this part in the new revised version.
- Your revision mentions:”Grain speed (formula 7) depends on the grain roughness. I pointed this out in my review for version 1. The authors did not respond to this comment.”It is true that our previous response to this issue was not clear enough, and the reason why we did not reflect this part of the revision in the revised draft is that we recalculated the Reynolds number for the granular phase based on your previous correction and found that the Reynolds number has changed significantly. In the new Reynolds number interval, the settling velocity that our particle phase fits into is a different equation that does not involve drag coefficient, so we did not make an explicit change on this issue. For this reason we have added more detail to this section by mentioning the issue of particle roughness in the revised version.
In summary, we have carefully considered and addressed all your comments and suggestions in the revised manuscript. We believe that these revisions have strengthened the clarity and accuracy of our work.
Thank you once again for your valuable feedback and constructive criticism. We remain committed to ensuring the quality and rigor of our research.
Best regards,
Mingwang Xiang
Reviewer 3 Report
Comments and Suggestions for Authors
Accept in present form
Comments on the Quality of English Language
Accept in present form
Author Response
Dear Reviewer,
We hope this message finds you well. We would like to express our sincere gratitude for your positive feedback and approval for the publication of our manuscript titled "Investigation of the effect of pumping depth and frequency of flapping hydrofoil on suspended matter discharge characteristics" Your acknowledgment of the improvements made to the manuscript is truly encouraging and reinforces our dedication to delivering high-quality research.
We have taken note of your suggestion regarding minor editorial editing, and we will ensure that any necessary adjustments are made to enhance the overall clarity and readability of the manuscript.
Your support and constructive feedback throughout the review process have been invaluable to us, and we greatly appreciate the time and effort you have invested in evaluating our work.
Once again, thank you for your positive assessment and for your confidence in the readiness of our manuscript for publication. We look forward to contributing our research findings to the scientific community through your esteemed journal.
Best regards,
Mingwang Xiang